# Minimal Geometry-Distortion Constraint for Unsupervised Image-to-Image Translation

## Abstract

Unsupervised image-to-image (I2I) translation, which aims to learn a domain mapping function without paired data, is very challenging because the function is highly under-constrained. Despite the significant progress in constraining the mapping function, current methods suffer from the *geometry distortion* problem: the geometry structure of the translated image is inconsistent with the input source image, which may cause the undesired distortions in the translated images. To remedy this issue, we propose a novel I2I translation constraint, called *Minimal Geometry-Distortion Constraint* (MGC), which promotes the consistency of geometry structures and reduce the unwanted distortions in translation by reducing the randomness of color transformation in the translation process. To facilitate estimation and maximization of MGC, we propose an approximate representation of mutual information called relative Squared-loss Mutual Information (rSMI) that can be efficiently estimated analytically. We demonstrate the effectiveness of our MGC by providing quantitative and qualitative comparisons with the state-of-the-art methods on several benchmark datasets.

## 1 Introduction

Image-to-image translation, or domain mapping, aims to translate an image in the source domain $\mathcal{X}$ to the target domain $\mathcal{Y}$. It has been extensively studied (Pathak et al., 2016; Isola et al., 2017; Liu et al., 2019) and has been applied to various vision tasks (Sela et al., 2017; Siddiquee et al., 2019; Ghosh et al., 2019; Tomei et al., 2019; Wu et al., 2019). Early works considered supervised image-to-image (I2I) translation, where paired samples $\{(x_i, y_i)\}_{i=1}^N$ drawn from the joint distribution $P_{XY}$ are available. In the presence of paired data, methods based on conditional generative adversarial networks can generate high-quality translations (Isola et al., 2017; Wang et al., 2018; Pathak et al., 2016). However, since paired data are often unavailable or expensive to obtain, unsupervised I2I translation has attracted intense attention in recent years (Zhu et al., 2017; Yi et al., 2017; Kim et al., 2017; Benaim & Wolf, 2017; Huang et al., 2018; Lee et al., 2019; Kim et al., 2019; Park et al., 2020).

Benefiting from generative adversarial networks (GANs) (Goodfellow et al., 2014), one can perform unsupervised I2I translation by finding $G_{XY}$ such that the translated images and target domain images have similar distributions, *i.e.*, $P_{G_{XY}(X)} \approx P_Y$. Due to an infinite number of functions that can satisfy the adversarial loss, GAN alone cannot guarantee the learning of the true mapping function, resulting in sub-optimal translation performance. To remedy this issue, various kinds of constraints have been placed on the learned mapping function. For instance, the well-known cycle-consistency (Zhu et al., 2017; Kim et al., 2017; Yi et al., 2017) enforces the translation function $G_{XY}$ to be bijective. DistanceGAN (Benaim & Wolf, 2017) preserves the pairwise distances in the source images. GcGAN (Fu et al., 2019) forces the function to be smooth w.r.t. certain geometric transformations of input images. DRIT++ (Lee et al., 2019) and MUNIT (Huang et al., 2018) learn disentangled representations by embedding images onto a domain-invariant content space and a domain-specific attribute space and the mapping function can be derived from representation learning components.

However, the mapping functions learned by current methods are still far from satisfactory in real applications. Here we consider a simple but widely applicable image translation task, *i.e.*, geometry-invariant translation task. In this task, the geometric structure (e.g. the shapes of objects) in images in the source and target domain is invariant and the variation of photometric information of a certain geometric area is expected to conform with the change of style information, such as the colour of a leaf is green in summer and white in winter. Existing methods enforced in geometry-invariant translation

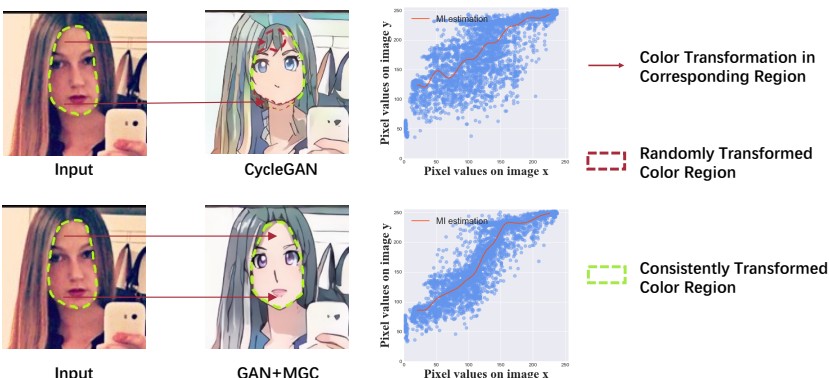

Figure 1: The illustration of how random color transformation causes the geometry-distortion problem in unsupervised image translation. Images in the first column are input images, and images in the second column are the translated images by CycleGAN and GAN+MGC. Visually, as the color transformation of the corresponding region between the input and translated image shows, the color of the human face is translated to several colors randomly by CycleGAN, leading to the distortion of face shape. In contrast, the color transformation in the GAN+MGC is consistent, and thus preserve the shape of the human face. To reveal the randomness of color transformation quantitatively, the third column images show that non-linear dependencies between pixel values in the input image and its corresponding pixel value in the translated image. Obviously, the geometry-preserved translated image (by GAN+MGC) has stronger color dependency than geometry-distorted one (by CycleGAN).

tasks still suffer from *geometry distortion* problem, where the geometry structures in the source and translated image are not consistent, resulting in the mismatch of input and translated images. A representative example is that the mapping function $G_{XY}$ learned with the cycle-consistency (Zhu et al., 2017) often changes the geometry structures of digits in the SVHN $\rightarrow$ MNIST translation task, so some digits in source domain images are translated accidentally into other digits. The *geometry distortion* problem hinders the application of unsupervised geometry-invariant translation methods into a wide range of computer vision applications, such as domain adaptation (Hoffman et al., 2017), segmentation (Zhu et al., 2017) and style transfer (Huang et al., 2018).

In this paper, we propose a new constraint for unsupervised geometry-invariant image translation, called *minimal geometry-distortion constraint* (MGC), as a general I2I translation constraint to guarantee the consistency of geometry structure of source and translated images, and thus reduce translation mismatch in the translation process. We observe that the pixel values before and after translation are usually highly correlated if the geometric structure is preserved because the color transformation is more regular within specific object regions. Taking the color transformation of a leaf as an example, the transformation of a green leaf into a red leaf contain less randomness than into a colorful one. Based on this observation, we propose a mutual information (MI)-based dependency measure that models the nonlinear relationships of pixel values in the source and translated images. To estimate MI from data, we propose the relative Squared-Loss Mutual Information (rSMI) which can be efficiently estimated in an analytic form. By maximizing rSMI together with the GAN loss, our approach can significantly reduce the geometry distortion by better preserving geometric structures. In the experiments, we incorporate our *minimal geometry-distortion constraint* into the GAN framework and show its effectiveness of preserving geometric structures when used both independently and combined with existing constraints (*e.g.* cycle consistency) to show its compatibility. The quantitative and qualitative comparisons with baselines (models without MGC) and state-of-the-art methods on several datasets demonstrate the superiority of the proposed MGC constraint.

## 2 RELATED WORK

**Unsupervised Image-to-Image Translation.** In unsupervised image-to-image (I2I) translation, unaligned examples drawn individually from the marginal distribution of the source domain and target domain are available. Although the subject has obtained some promising progress in recent years, only several works study it from an optimization perspective. Specifically, Cyclic consistency based GAN, *e.g.,* CycleGAN (Zhu et al., 2017), DualGAN (Yi et al., 2017) and DiscoGAN (Kim et al., 2017), is a general approach for this problem. DistanceGAN (Benaim & Wolf, 2017) and GcGAN (Fu et al., 2019) further introduced distance and geometry transformation consistency to

constraint the search space of mapping functions. Instead of exploiting general constraints for the subject, more works developed novel frameworks to investigate special settings of unsupervised I2I translation. Several other works (Huang et al., 2018; Choi et al., 2018; Lee et al., 2019; 2018; Shen et al., 2019) mapped the content and style information of images into disentangled spaces for multi-modal translations. However, we find that the complex neural networks and many hyper-parameters make the optimization process unstable (Kim et al., 2019). In addition, (Dosovitskiy & Brox, 2016; Johnson et al., 2016; Mechrez et al., 2018; Katzir et al., 2019) tried to reduce the perceptual loss or content loss based on a pre-trained VGG model to reduce the content of two domain image, which is computationally cost and cannot be easily adapted to the data on hand. Further, Wu et al. (2019); Katzir et al. (2019) give efforts to change the geometry of source images using geometry information extracted by cVAE (Esser et al., 2018; Kingma & Welling, 2013) or pre-trained VGG network. However, for many image translation applications that need the geometry structures preservation, *e.g.* domain adaptation, unsupervised segmentation, geometry-invariant image translation, our method provides a simple way to achieve the goal of preserving geometry-structure.

**Mutual Information (MI).** Mutual information is a fundamental measure of dependency between two random variables, and it is widely used in machine learning and is particularly suitable for canonical tasks such as multi-modalities images registration (Zitova & Flusser, 2003; Maes et al., 1997; Luan et al., 2008). Since computing MI is difficult (Paninski, 2003), researchers have taken much effort to improve the estimation of MI. For example, early works studied Non-parametric models based on Kernel Density Estimator (KDE) (Jonsson & Felsberg, 2006; Krishnamurthy et al., 2014; Kandasamy et al., 2015; Singh & Póczos, 2014a;b), K-nearest Neighbor Method (KNN) (Kraskov et al., 2004; Kozachenko & Leonenko, 1987), and likelihood-ratio estimator (Suzuki et al., 2008) for MI estimation. Subsequent works improved the performance in more complicated cases such as discrete-continuous mixtures (Moon et al., 2017; Gao et al., 2017) and segmentation (Zhao et al., 2019). Recently, MINE (Belghazi et al., 2018; Hjelm et al., 2018) showed that the estimation of mutual information between high dimensional continuous random variables can be achieved by gradient descent over neural networks.

## 3 METHODOLOGY

Unsupervised I2I translation aims to find a mapping function $G_{XY}$ between two domains $\mathcal{X}$ and $\mathcal{Y}$ given unpaired samples $\{x_i\}_{i=1}^N$ and $\{y_j\}_{j=1}^M$ drawn from the marginal distributions $P_X$ and $P_Y$, respectively. To solve *geometry-distortion* problem, we first present our motivation of placing the MI-based minimal geometry-distortion constraint, and then give the details about our proposed minimal geometry-distortion constraint (MGC), which aims to reduce the randomness of color transform in the translation process and thus promote the consistency of geometry structure between source and translated images.

### 3.1 MOTIVATION

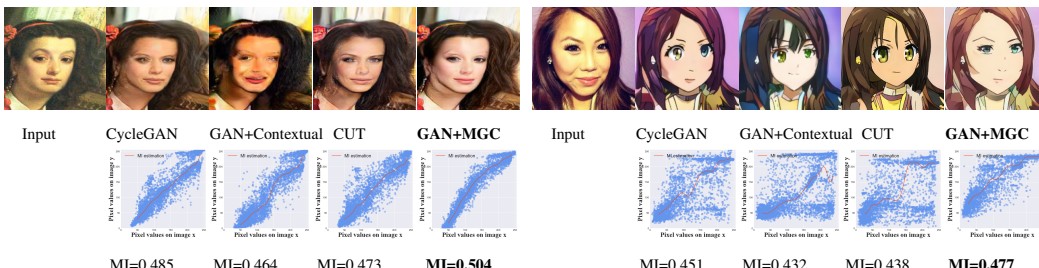

Figure 2: Unsupervised image translation examples on Portrait → Photo, Selfie → Anime. The top row is the translated results by each method. The bottom row is the scatter plot of the pixel values in the input image $x$ and its corresponding pixel value in the translated image $\hat{y}$, which shows the non-linear dependency of pixel values in two images. Obviously, the stronger the dependency between pixel values in the input image (X-axis) and the translated images (Y-axis), the better the geometry structure of the input image is maintained. MI stands for mutual information, which is estimated by our rSMI method.

As illustrated in Figure 2, 4 (a), and 5, advanced methods (*e.g.* CycleGAN, CUT (Park et al., 2020), Contexual loss (Mechrez et al., 2018), U-GAT-IT (Kim et al., 2019), MUNIT (Huang et al., 2018) can change the geometry structure of input images and potentially cause the mismatch between input and translated images. Therefore, it is essential to enforce a constraint such that we can ensure the learned function $G_{XY}$ change the image style with minimal geometry distortion. Our work is the first to explore such constraints for unsupervised image-to-image translation.

To reveal the reason for the *geometry-distortion* in I2I translation, we plot the corresponding pixel values of images before and after translation at the bottom row of Figure 2. We can see that the pixel values in the translated image (Y-axis) are less dependent of the pixel values (X-axis) in the input images using previous methods (*e.g.* CycleGAN, CUT, Contextual loss). As such, one color within an object region can be randomly mapped to various colors after translation. A geometric area in a single image is normally occupied by a certain color, so the randomness of the color transformation in the translation process using previous methods distorts the geometry structure of the input image. Therefore, reducing the randomness of color transformation is a way to alleviate the *geometry-distortion* problem in current I2I translation methods.

Motivated by the analysis, we develop the *minimal geometry-distortion constraint* (MGC) as a general and effective constraint to preserve the pixel-level structure during the translation process. MGC exploits mutual information to model the non-linear dependencies of pixel values between the input and translated images, thus reducing the randomness of color transformation in the translation. As illustrated in Figure 3, our MGC is enforced into the input and translated images and thus allows one-sided unsupervised domain mapping, i.e., $G_{XY}$ can be trained independently from $G_{YX}$. Applying our MGC to a vanilla GAN, the pixel values before and after translation have stronger dependency (higher MI), and thus better preserves the geometric structures as shown in Figure 2. In the following, we present the details of our approach.

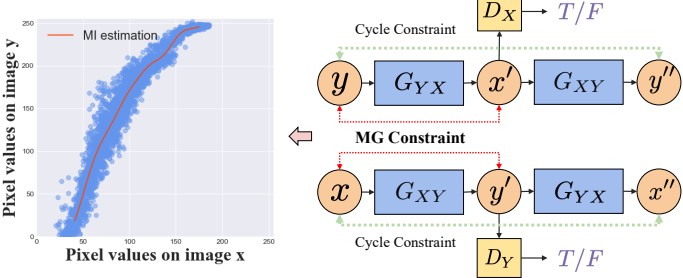

Figure 3: An illustration of minimal geometry-distortion constraint. The left figure shows that the pixel value in the input image $x$ and its corresponding pixel value in the translated image $\hat{y}$ have strong non-linear dependencies, and so we add the MG constraint to model the dependencies of pixel values in two domain images as the right figure shows, and thus preserve the image geometry in translation. This constraint is also compatible with other constraints, such as cycle-constraint. The pixel dependency example is from portrait$\rightarrow$ photo dataset.

## 3.2 APPROXIMATE REPRESENTATION OF MUTUAL INFORMATION

For a source domain image $x_i \in \mathcal{X}$ and its translation $\hat{y}_i = G_{XY}(x_i)$, we denote $V^{x_i}$ and $V^{\hat{y}_i}$ as the random variables for pixels in $x_i$ and $\hat{y}_i$, respectively. Thus, pixels in $x_i$, i.e., $\{v_j^{x_i}\}_{j=1}^M$, can be regarded as data sampled from $P_{V^{x_i}}$, and the pixels in $\hat{y}_i$, i.e., $\{v_j^{\hat{y}_i}\}_{j=1}^M$, can be considered as data sampled from $P_{V^{\hat{y}_i}}$, where $M$ is the number of pixels of the image. Formally, the mutual information between $V^{x_i}$ and $V^{\hat{y}_i}$ is

$$MI(V^{x_i}, V^{\hat{y}_i}) = \mathbb{E}_{(v^{x_i}, v^{\hat{y}_i}) \sim P_{(V^{x_i}, V^{\hat{y}_i})}} \left( \log \frac{P_{(V^{x_i}, V^{\hat{y}_i})}}{P_{V^{x_i}} \otimes P_{V^{\hat{y}_i}}} \right) \qquad (1)$$

where $P_{(V^{x_i}, V^{\hat{y}_i})}$ is the joint distribution of $V^{x_i}$ and $V^{\hat{y}_i}$, $P_{V^{x_i}} \otimes P_{V^{\hat{y}_i}}$ is the product of two marginal distributions $P_{V^{x_i}}$ and $P_{V^{\hat{y}_i}}$. Because $V^{x_i}$ and $V^{\hat{y}_i}$ are low-dimensional, a straightforward way to estimate (1) is to estimate the distributions $P$ based on the histogram of the images. Next, we will introduce how we estimate the mutual information between pixels from two domain images and backpropagate it to optimize parameters in the translation network.

To enable efficient backpropagation, we propose the relative Squared-loss Mutual Information (rSMI), which is an extension of the well-known Squared-loss Mutual Information (SMI) (Suzuki et al., 2009). For conventional presentation, we denote $P_{V^{x_i}} \otimes P_{V^{\hat{y}_i}}$ as $S_i$, and $P_{(V^{x_i}, V^{\hat{y}_i})}$ as $Q_i$. Then, the SMI based on Pearson (PE) Divergence (Sugiyama et al., 2012) between $P_{V^{x_i}}$ and $P_{V^{\hat{y}_i}}$ is expressed as:

$$SMI(V^{x_i}, V^{\hat{y}_i}) = D_{PE}(P_{V^{x_i}} \otimes P_{V^{\hat{y}_i}} || P_{(V^{x_i}, V^{\hat{y}_i})}) = D_{PE}(S_i || Q_i) = \mathbb{E}_{Q_i}[(\frac{S_i}{Q_i} - 1)^2]. \quad (2)$$

Because $\frac{S_i}{Q_i}$ is unbounded, $SMI(V^{x_i}, V^{\hat{y}_i})$ can be infinity, causing numeric instability in the back-propagation. We thus use the relative Pearson(rPE) Divergence (Yamada et al., 2013) to alleviate the problem:

$$D_{rPE}(S_i || Q_i) = D_{PE}(S_i || \beta S_i + (1 - \beta)Q_i). \quad (3)$$

Here, we introduce the mixture distribution $\beta S_i + (1 - \beta)Q_i$, $\beta \in (0, 1)$, to replace $Q_i$. Benefiting from the modification, the density ratio will be bounded to $[0, \frac{1}{\beta}]$. Thus, the proposed rSMI between $V^{x_i}$ and $V^{\hat{y}_i}$ can be written as:

$$rSMI(V^{x_i}, V^{\hat{y}_i}) = D_{rPE}(P_{V^{x_i}} \otimes P_{V^{\hat{y}_i}} || P_{(V^{x_i}, V^{\hat{y}_i})}) = \mathbb{E}_{\beta S_i + (1-\beta)Q_i}[(\frac{S_i}{\beta S_i + (1 - \beta)Q_i} - 1)^2]. \quad (4)$$

To estimate the $rSMI(V^{x_i}, V^{\hat{y}_i})$, we directly estimate the density ratio using a linear combination of kernel functions of $\{v_j^{x_i}\}_{j=1}^M$ and $\{v_j^{\hat{y}_i}\}_{j=1}^M$:

$$\frac{S_i}{\beta S_i + (1 - \beta)Q_i} = \omega_\alpha(v^{x_i}, v^{\hat{y}_i}) = \sum_{l=1}^m \alpha_l \phi_l(v^{x_i}, v^{\hat{y}_i}) = \alpha^T \phi(v^{x_i}, v^{\hat{y}_i}),$$

where $\phi \in \mathbb{R}^m$ is the kernel function, $\alpha \in \mathbb{R}^m$ is the parameter vector we need to solve, and $m$ is the number of kernels. Referring to the least-squares density-difference estimation (Sugiyama et al., 2013), the solved optimal solution of $\hat{\alpha}$ is (the derivation is given in the appendix A.1 ):

$$\hat{\alpha} = (\hat{H} + \lambda R)^{-1}\hat{h},$$

$$\hat{H} = \frac{1 - \beta}{n}(K \circ L)(K \circ L)^T + \frac{\beta}{n^2}(KK^T) \circ (LL^T), \qquad \hat{h} = \frac{1}{n^2}(K1_n) \circ (L1_n),$$

where $R$ is a positive semi-definite regularization matrix, $n$ is the sample number, $1_n$ is the n-dimensional vector filled by ones, and $K$ and $L$ are two $m \times n$ matrices composed by kernel functions, and the Hadamard product of $K$ and $L$ is used to define $\phi$, that is $\phi(v^{x_i}, v^{\hat{y}_i}) = K(v^{x_i}) \circ L(v^{\hat{y}_i})$. Finally, an appropriate mutual information estimator of with smaller bias is expressed as:

$$\widehat{rSMI}(V^{x_i}, V^{\hat{y}_i}) = 2\hat{\alpha}^T\hat{h} - \hat{\alpha}^T\hat{H}\hat{\alpha} - 1. \quad (5)$$

Note that, the computation of $\widehat{rSMI}(V^{x_i}, V^{\hat{y}_i})$ is resource friendly, as it can be solved analytically. Thus, the parameters in the translation neural network can be efficiently updated by backprogation.

## 3.3 FULL OBJECTIVE

Following the analysis above, our minimal geometry-distortion constraint (MGC) for I2I translation using mutual information can be expressed as:

$$\mathcal{L}_{mgc} = \frac{1}{N} \sum_{i=1}^N \widehat{rSMI}(V^{x_i}, V^{\hat{y}_i}) = \frac{1}{N} \sum_{i=1}^N \widehat{rSMI}(V^{x_i}, V^{G_{XY}(x_i)}), \quad (6)$$

where $N$ is the number of samples. We directly maximize $\mathcal{L}_{mgc}$ to guarantee more local geometric structures of images being invariant in the translation process. By combining our MGC with the standard adversarial loss, the image geometry will be preserved while its style is changed. As a result, one-sided unsupervised domain mapping can be targeted. The full objective will take the form:

$$\min_{G_{XY}} \max_{D_Y} \mathcal{L}_{gan+mgc}(G_{XY}, D_Y) = \mathcal{L}_{gan}(G_{XY}, D_Y) - \lambda_{mgc}\mathcal{L}_{mgc}(G_{XY}), \quad (7)$$

where $\mathcal{L}_{gan}$ is the adversarial loss (Goodfellow et al., 2014), which introduced a discriminator $D_Y$, to encourage the distribution of output matches the distributions of target domain images, *i.e* $P_{G_{XY}(X)} \approx P_Y$. The objective function as follows:

$$\mathcal{L}_{gan}(G_{XY}, D_Y) = \mathbb{E}_{y \sim P_Y}[\log D_Y(y)] + \mathbb{E}_{x \sim P_X}[\log(1 - D_Y(G_{XY}(x)))]. \quad (8)$$

In Equation 7, $\lambda_{mgc}$ is a hyperparameter to weight $\mathcal{L}_{gan}$ and $\mathcal{L}_{mgc}$ in the training procedure. The proposed MGC can be integrated into various I2I translation frameworks, *e.g.,* CycleGAN (Zhu et al., 2017) and CUT(Park et al., 2020), by replacing the loss $\mathcal{L}_{gan}$ with the losses in these methods.

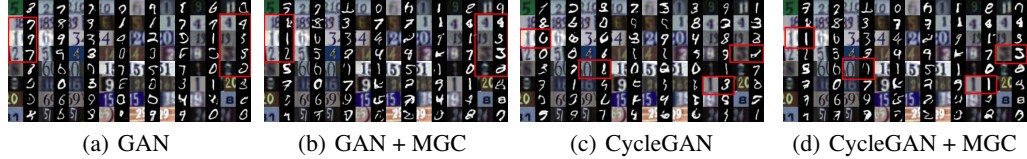

| (a) GAN | (b) GAN + MGC | (c) CycleGAN | (d) CycleGAN + MGC |

Figure 4: Qualitative comparisons on SVHN→MNIST. From Figure (a) and (b), we can see that the GAN method has no collapse solution by combining with our MGC. Also, the geometry distortion problem in CycleGAN is alleviated after incorporating with MGC.

## 4 EXPERIMENTS

In this section, we perform experiments on three typical unsupervised geometry-invariant image translation applications: Digits Translation, Unsupervised Segmentation and Image Generation (*e.g.* Cityscapes (Cordts et al., 2016) ), and Style Transfer. To show the effectiveness of our MGC, we couple our minimal geometry-distortion constraint (MGC) with the vanilla GAN. Further, we incorporate MGC with some popular methods such as cycle-consistency (Zhu et al., 2017), GcGAN(Fu et al., 2019), and U-GAT-IT (Kim et al., 2019) to show its compatibility with other constraints. Then we make qualitative and quantitative comparisons with the recent published unsupervised I2I translation methods *e.g.* CycleGAN, GcGAN, CoGAN (Liu & Tuzel, 2016), SimGAN (Shrivastava et al., 2017), BiGAN (Donahue et al., 2016) , DistanceGAN (Benaim & Wolf, 2017), CUT (Park et al., 2020)), the Contextual Loss (Mechrez et al., 2018), DRIT++ (Lee et al., 2019), UNIT (Liu et al., 2017), MUNIT (Huang et al., 2018), AGGAN (Tang et al., 2019), and U-GAT-IT (Kim et al., 2019). However, the current baselines have their own advantages and disadvantages: some baselines perform well on one task but perform poorly on other tasks. For example, some style transfer methods do not perform well on unsupervised image segmentation. As such, following the current literature, we compare our methods with SOTA methods for each application. At last, we perform the ablation study by varying the hyper-parameter $\lambda_{mgc}$. In the appendix, we firstly explore the use of our MGC on the geometry-variant dataset, *i.e.*, cat2dog dataset (Lee et al., 2019) (the analysis is given in the A.5), and then investigate the influence of our MGC on the generation diversity A.2.2 and training stability A.2.3. We examine all the experiments three times and report the average scores to reduce random errors.

For the implementation of the mutual information estimator presented in section 3.2, we set the hyperparameter $\beta$ to 0.5 (more analysis and experiments about other values of $\beta$ are given at the appendix A.2.1), and utilize nine Gaussian kernels for both input images $x$ and translated images $\hat{y}$. Then we apply our MGC to all the baselines and keep other experimental details including hyper-parameters, networks in baselines the same. Due to page limit, we provide more experimental details and qualitative results in the Appendix A.4 and A.7, respectively.

Table 1: Classification accuracy for digits experiments.

| | Translated Images as Test set | | | Translated Images as Training set | | |
|---|---|---|---|---|---|---|
| Method | S → M | M → M-M | M-M → M | S → M | M → M-M | M-M → M |
| GAN alone | 21.3±9.5 | 54.6±40.5 | 80.3±3.5 | 28.6±10.8 | 45.7±31.2 | 95.5±0.4 |
| **GAN + MGC** | 37.3±1.2 | 96.3±0.2 | 90.9±0.5 | 47.9±2.3 | 86.2±1.9 | 96.0±0.1 |
| CycleGAN | 26.1±8.1 | 95.3±0.4 | 84.7±2.5 | 31.6±5.6 | 83.8±3.0 | 95.9±0.4 |
| **CycleGAN + MGC** | 38.0±0.5 | **96.7±0.1** | 91.5±0.3 | 47.4±2.0 | 87.7±2.1 | **96.1±0.2** |
| GcGAN-*rot* | 32.5±2.0 | 95.0±0.6 | 85.9±0.8 | 40.9±6.5 | 84.6±2.8 | 96.0±0.1 |
| **GcGAN-*rot* + MGC** | 36.5±1.3 | 96.4±0.3 | 91.8 ±1.0 | 47.5±1.2 | 89.5±0.6 | 96.1±0.1 |
| GcGAN -*vf* | 33.3±4.2 | 95.2±0.4 | 84.5±1.5 | 31.6±5.6 | 83.8±3.0 | 95.9±0.4 |
| **GcGAN-*vf* + MGC** | 37.0±0.8 | 96.6±0.3 | 91.8±0.8 | 49.5±4.9 | 87.8±2.3 | 96.0±0.1 |
| **CycleGAN + rot + MGC** | 39.0±0.5 | 96.5±0.3 | 91.8±1.0 | 50.5±1.8 | **89.8±0.5** | 96.1±0.1 |
| **CycleGAN + vf + MGC** | **44.6±6.8** | **96.7±0.3** | **92.0±0.8** | **51.3±5.4** | 89.0±0.8 | 96.1±0.1 |

### 4.1 DIGIT TRANSLATION

Following (Fu et al., 2019; Benaim & Wolf, 2017), we examine three digit I2I translation tasks: SVHN→MNIST, MNIST-M→MNIST and MNIST→MNIST-M [1]. The models are trained on the

---
[1]refer to S→M, M-M→M and M→M-M

training split with images size $32 \times 32$, and $\lambda_{mgc}$ is set to 20. We adopt the classification accuracy as the evaluation metric, and design two evaluation methods: (1) we train a classifier on the target dataset's training split. The fake images translated from the source dataset's test images are used to compute the classification accuracy. This evaluation method can only measure the quality of translated images. (2) a classifier is trained on the translated images from the source dataset's training images, and test the performance of this classifier on the target dataset's test split. This evaluation method can measure both the quality and diversity of translation images, but it is unstable [2].

We conduct each experiment five times to reduce the randomness of GAN-based approaches. The scores are reported in Table 1. Generally, by incorporating our MGC, all the baselines show promising improvements in both accuracy and stability, especially for the challenging task S→M. Some qualitative results are shown in Figure 4. More details and results are given in Appendix A.4.1 and A.7.4, respectively.

## 4.2 CITYSCAPES

Following (Fu et al., 2019; Zhu et al., 2017), we train the models using the unaligned 3975 images of Cityscapes (Cordts et al., 2016) with $128 \times 128$ resolution. We evaluate the domain mappers using FCN scores and scene parsing metrics as previously done in (Zhu et al., 2017). Specifically, for parsing→image, we use the pre-trained FCN-8s (Long et al., 2015) provided by pix2pix (Isola et al., 2017) to predict the segmentation label maps from translated images, then compare them against the ground truth labels using parsing metrics including pixel accuracy, class accuracy, and mean IoU. For image→parsing, the translated label maps are also compared against the ground truth.

Table 2: Parsing scores on Cityscapes. The scores with [*] are reproduced on a single GPU using the codes provided by the authors. Qualitative results are given at the Appendix A.7.2.

| Methods | image → parsing | | | parsing → image | | |
|---|---|---|---|---|---|---|
| | pixel acc | class acc | mean IoU | pixel acc | class acc | mean IoU |
| CoGAN | 0.45 | 0.11 | 0.08 | 0.40 | 0.10 | 0.06 |
| BiGAN/ALI | 0.41 | 0.13 | 0.07 | 0.19 | 0.06 | 0.02 |
| SimGAN | 0.47 | 0.11 | 0.07 | 0.20 | 0.10 | 0.04 |
| DistanceGAN | - | - | - | 0.53 | 0.19 | 0.11 |
| GcGAN-rot [*] | 0.574 | 0.176 | 0.132 | 0.551 | 0.197 | 0.129 |
| **GcGAN-rot + MGC** | 0.579 | 0.180 | 0.136 | 0.622 | 0.215 | 0.152 |
| CycleGAN [*] | 0.58 | 0.203 | 0.152 | 0.52 | 0.17 | 0.11 |
| **CycleGAN + MGC** | **0.591** | **0.208** | **0.156** | 0.540 | 0.185 | 0.127 |
| CUT [*] | \ | \ | \ | 0.695 | 0.259 | 0.178 |
| **CUT + MGC** | \ | \ | \ | **0.699** | **0.263** | **0.182** |

As reported in Table 2, the results of all the image translation methods are improved if further constrained by our MGC. In particular, GcGAN coupled with MGC yields a promising improvement compared with GcGAN in the parsing → image task.

## 4.3 STYLE TRANSFER

We implement the style transfer task on anime2selfie (Kim et al., 2019), horse2zebra (Zhu et al., 2017), photo2portrait (Lee et al., 2018). We choose CycleGAN, GcGAN, AGGAN, DRIT, UNIT, MUNIT, and CUT as baselines. All images are resized to $256 \times 256$ resolution and $\lambda_{mgc}$ is set to 5 for all experiments. More details are given in A.4.4.

Following the recent work (Kim et al., 2019), we use KID score (Bińkowski et al., 2018) as the evaluation metric. The results are reported in the Table 3, we can see that the vanilla GAN method coupled with our MGC can achieve the comparable results with those methods with larger model size. In addition, a simple generator based on res-blocks trained by the combination of cycle, geometry and our MGC constraint can achieve SOTA performance on almost all datasets. As the qualitative results shown in Figure 5, after adding our MGC, the translated images retain the geometric structure of the original image, and is consistent with the style of the target image. More results are given in the appendix A.7.3, and the light version of U-GAT-IT with our MGC can achieve better performance of full version of U-GAT-IT, at the situation of saving a half size of parameters. Then we conducted a

---

[2]Note that, the setting of I2I translation is different from domain adaptation. The latter one has access to the labels of source domain images while the former does not.

Table 3: KID scores for style transfer tasks. The results of baselines (AGGAN (Tang et al., 2019) , DRIT (Lee et al., 2019) , UNIT (Liu et al., 2017) , MUNIT (Huang et al., 2018)) are from (Kim et al., 2019). Here U (light) is the light version of U-GAT-IT.

| | Params | selfie2anime | horse2zebra | photo2por | anime2selfie | zebra2horse | por2photo |
|---|---|---|---|---|---|---|---|
| AGGAN | \ | 14.63±0.55 | 7.58±0.71 | 2.33±0.36 | 12.72±1.03 | 8.80±0.66 | 2.19±0.40 |
| DRIT | 65.0M | 15.08±0.62 | 9.79±0.62 | 5.85±0.54 | 14.85±0.60 | 10.98±0.55 | 4.76±0.72 |
| UNIT | \ | 14.71±0.59 | 10.44±0.67 | **1.20±0.31** | 26.32±0.92 | 14.93±0.75 | 1.42±0.24 |
| MUNIT | 46.6M | 13.85±0.41 | 11.41±0.83 | 4.75±0.52 | 13.94±0.72 | 16.47±1.04 | 3.30±0.47 |
| U-GAT-IT(full) | 134.0M | 11.61±0.57 | 7.06±0.8 | 1.79±0.34 | 11.52±0.57 | 7.47±0.71 | 1.69±0.53 |
| U-GAT-IT(light) | 74.0M | 12.31±0.50 | 7.25±0.8 | 3.43±0.28 | 15.22±0.51 | 9.39±0.48 | 2.67±0.33 |
| **U (light)+MGC** | 74.0M | **10.37±0.32** | 5.19±0.46 | 3.19±0.26 | **10.30±0.47** | 7.80±0.48 | 2.18±0.26 |
| GAN+Contextual | 588.1M | 12.77±0.38 | 9.39±0.39 | 3.95±0.26 | 14.81±0.41 | 10.36±0.51 | 3.05±0.25 |
| **GAN + MGC** | 14.1M | 11.37±0.41 | 7.28±0.52 | 3.86±0.39 | 11.61±0.40 | 7.15±0.46 | 1.58±0.25 |
| CycleGAN | 28.3M | 13.08±0.49 | 8.05±0.72 | 1.84±0.34 | 11.84±0.74 | 8.0±0.66 | 1.82±0.36 |
| **Cycle + MGC** | 28.3M | 11.66±0.41 | 6.59±0.49 | 2.91±0.22 | 10.83±0.44 | 6.77±0.52 | 1.62±0.15 |
| GcGAN | 16.9M | 11.89±0.42 | 7.05±0.45 | 2.24±0.26 | 13.28±0.35 | 7.67±0.47 | 1.84±0.28 |
| **GcGAN + MGC** | 16.9M | 10.75±0.42 | 5.12±0.44 | 1.97±0.24 | 10.96±0.40 | 7.10±0.50 | 1.64±0.22 |
| CUT | 18.1M | 12.1±0.42 | 8.45±0.45 | 2.85±0.33 | 12.45±0.54 | 8.99±0.5 | 2.23±0.31 |
| **CUT + MGC** | 18.1M | 11.75±0.41 | 6.26±0.44 | 2.31±0.3 | 12.05±0.44 | 8.4±0.43 | 2.11±0.26 |
| **Gc+Cycle+MGC** | 45.2M | 10.61±0.44 | **4.82±0.68** | 1.64±0.24 | 10.92±0.35 | **6.28±0.52** | **1.31±0.27** |

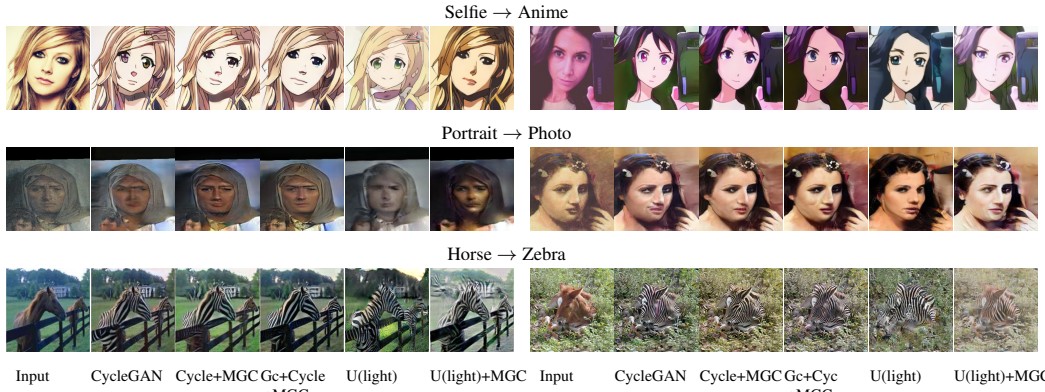

Selfie → Anime

Portrait → Photo

Horse → Zebra

Input    CycleGAN    Cycle+MGC Gc+Cycle   U(light)    U(light)+MGC    Input    CycleGAN    Cycle+MGC Gc+Cyc    U(light)    U(light)+MGC
                                +MGC                                                                      +MGC

Figure 5: Qualitative results on style transfer datasets, including Selfie → Anime, Portrait → Photo, Horse → Zebra. More qualitative results are given in A.7.3. It can be seen that the face shape is better preserved by the translation model empowered by our MGC.

Table 4: The results of User Study: the percentage of users prefer a particular model. To avoid the concern of cherry-picking, qualitative results of U-GAT-IT and our results are used as the evaluation images in the user study. Sample images are given in Appendix A.4.4.

| | **Cyc+Gc+MGC** | U-GAT-IT | MUNIT | DRIT | CycleGAN |
|---|---|---|---|---|---|
| horse2zebra | **33.20** | 32.22 | 1.25 | 5.28 | 28.05 |
| selfie2anime | **47.85** | 37.22 | 1.67 | 2.94 | 10.32 |
| photo2portrait | **56.89** | 19.00 | 8.44 | 3.00 | 12.67 |
| Paramaters | 45.2MB | 134.0MB | 46.6MB | 65.0MB | 28.3MB |

user study, in which 180 participants were asked to choose the best translated image given the domain names *e.g.* selfie → anime, exemplar images in the source and target domains, and the corresponding translated images from different methods. The results shown in Table 4 demonstrate that most users choose the outputs of our method.

## 4.4 SENSITIVITY ANALYSIS

We study the influence of MGC by performing experiments with different $\lambda_{mgc}$. As shown in Table 5 and Figure 6, although the best $\lambda_{mgc}$ varies in each task, the performance of translation models are all improved to some extend after incorporating our MGC. However, when $\lambda_{mgc}$ becomes too large, the improvement with our MGC is limited as the model focuses on reducing geometry distortion and ignores the style information learned from GAN. More examples are given in the Appendix

Table 5: Sensitivity Analysis: the KID scores for different $\lambda_{mgc}$ of the model CycleGAN + MGC in the datasets horse2zebra and selfie2anime.

| $\lambda_{mgc}$ | 0 | 1 | 3 | 5 | 7 | 9 |
|---|---|---|---|---|---|---|
| horse2zebra | 8.05 | 7.28 | 6.94 | **6.59** | 6.73 | 6.75 |
| zebra2horse | 8.0 | 6.89 | **6.53** | 6.77 | 6.69 | 6.81 |
| selfie2anime | 13.08 | 12.52 | 11.68 | 11.66 | **11.26** | 11.37 |
| anime2selfie | 11.84 | 10.97 | 11.61 | 10.83 | **10.72** | 10.81 |

|  | MI=0.381 | MI=0.392 | MI=0.402 | MI=0.406 | MI=0.408 | MI=0.408 |
|---|---|---|---|---|---|---|
|  | MI=0.431 | MI=0.449 | MI=0.453 | MI=0.450 | MI=0.462 | MI=0.466 |
| Input | CycleGAN | $\lambda_{mgc} = 1$ | $\lambda_{mgc} = 3$ | $\lambda_{mgc} = 5$ | $\lambda_{mgc} = 7$ | $\lambda_{mgc} = 9$ |

Figure 6: Sensitivity analysis examples on Selfie $\rightarrow$ Anime. Obviously, the geometry distortion problem in CycleGAN is alleviated after incorporating with our MGC.

A.7.5. A practical strategy of choosing $\lambda_{mgc}$ is to find the largest $\lambda_{mgc}$ with normal style information using binary search. Specifically the first value of $\lambda_{mgc}$ can be set 5, which is suitable for most style transfer datasets.

## 5 CONCLUSION

In this paper, we propose the minimal geometry-distortion constraint (MGC) to improve geometry invariance in pixel-wise level for unsupervised image-to-image translation. In addition, we propose an expression of mutual information called relative Squared-loss Mutual Information(rSMI) with an analytical method for estimation. We evaluate our model quantitatively in a wide range of applications. The experimental results demonstrate that MGC achieves high quality translation to maintain the geometry of images in original domain.

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

# A APPENDIX

## A.1 DETAILS OF SOLVING $\alpha$

To solve the $rSMI(V^{x_i}, V^{\hat{y}_i})$, we directly estimate the density ratio using a linear combination of kernel functions of $\{v_j^{x_i}\}_{j=1}^M \in V^{x_i}$ and $\{v_j^{\hat{y}_i}\}_{j=1}^M \in V^{\hat{y}_i}$:

$$\frac{S_i}{\beta S_i + (1-\beta)Q_i} = \omega_\alpha(v^{x_i}, v^{\hat{y}_i}) = \sum_{l=1}^m \alpha_l \phi_l(v^{x_i}, v^{\hat{y}_i}) = \alpha^T \phi(v^{x_i}, v^{\hat{y}_i}), \tag{9}$$

where $\phi \in \mathbb{R}^m$ is the kernel function, $\alpha \in \mathbb{R}^m$ is the parameter vector we need to solve, and $m$ is the number of kernels. $\alpha$ is learned so that the following squared error $J(\alpha)$ (Sugiyama, 2013) is minimized:

$$J(\alpha) = \mathbb{E}_{\beta S + (1-\beta)Q_i}[(\omega_\alpha(v^{x_i}, v^{\hat{y}_i}) - \omega^*(v^{x_i}, v^{\hat{y}_i}))^2] = \mathbb{E}_Q[(1-\beta)\omega_\alpha^2] + \mathbb{E}_S[\beta\omega_\alpha^2 - 2\omega_\alpha] + J_0,$$

where $J_0$ is a constant number respect to $\alpha$, and therefore can be safely ignored. Thus, the optimization problem is given as:

$$\min_\alpha [\alpha^T H \alpha - 2\alpha^T h],$$

where

$$H = (1-\beta)\mathbb{E}_Q[\phi\phi^T] + \beta\mathbb{E}_S[\phi\phi^T], \qquad h = \mathbb{E}_S[\phi].$$

For computational efficiency, we define the kernel function $\phi(v^{x_i}, v^{\hat{y}_i})$ as the product of $K(v^{x_i}; k_c) \in \mathbb{R}^m$ and $L(v^{x_i}; l_c) \in \mathbb{R}^m$, which are kernel functions of $v^{x_i}$ and $v^{\hat{y}_i}$ respectively:

$$\phi(v^{x_i}, v^{\hat{y}_i}) = K(v^{x_i}) \circ L(v^{\hat{y}_i}),$$

where $\circ$ denotes the Hadamard product. Approximating the expectations in $H$ and $h$ by empirical averages, and adding a quadratic regularizer $\alpha^T R \alpha$ to avoid over-fitting, the objective function in our optimize problem becomes:

$$\hat{J}(\alpha) = [\alpha^T \hat{H} \alpha - 2\hat{h}^T \alpha + \lambda \alpha^T R \alpha], \tag{10}$$

where $R$ is the positive semi-definite regularization matrix, and

$$\hat{H} = \frac{1-\beta}{n}(K \circ L)(K \circ L)^T + \frac{\beta}{n^2}(KK^T) \circ (LL^T), \qquad \hat{h} = \frac{1}{n^2}(K1_n) \circ (L1_n),$$

where $n$ is the number of samples, $1_n$ is the n-dimensional vector filled by ones, and $K$ and $L$ are two $m \times n$ matrices composed by kernel functions. The equation 10 is a unconstrained quadratic problem, and thus could be solved by analytically and the optimal solution of $\hat{\alpha}$ is:

$$\hat{\alpha} = (\hat{H} + \lambda R)^{-1}\hat{h}.$$

## A.2 EXPERIMENTAL ANALYSIS

### A.2.1 $\beta$ ANALYSIS

We conduct the sensitive analysis of $\beta$ on the digits datasets (each experiment is repeated 3 times) and the results are shown as Figure 7 (b). We can see the performance of translation models are all improved with varied $\beta$, and we use 0.5 for convenience.

### A.2.2 GENERATION DIVERSITY ANALYSIS

We conduct the generation diversity experiments on the edge2shoes dataset. Following MUNIT (Huang et al., 2018), we calculate the average LPIPS distance between 1900 pairs of randomly generated images (sampled from 100 input images). MUNIT with MGC has the average LPIPS of 0.120, improving the diversity of original MUNIT model with 0.104 LPIPS score. Therefore, our MGC has no negative impact on generation diversity. Some generation examples are given as Figure 8.

### A.2.3 STABILITY ANALYSIS

We conduct the training stability analysis of our MGC on the digits datasets and the results are shown as Figure 7 (a). We can see the training procedure is stable with our MGC.

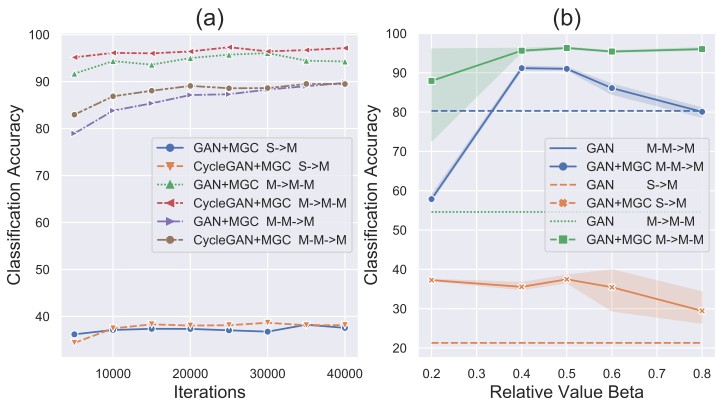

Figure 7: The training curves and the sensitive analysis about $\beta$ on Digits datasets

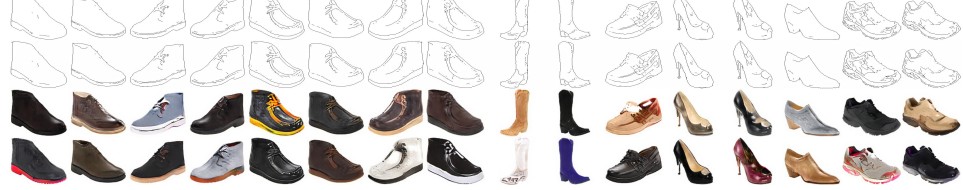

Figure 8: The generation example of MUNIT+MGC on the edge2shoes. Specifically, images at first two rows are source domain images and the others are translated images by MUNIT+MGC.

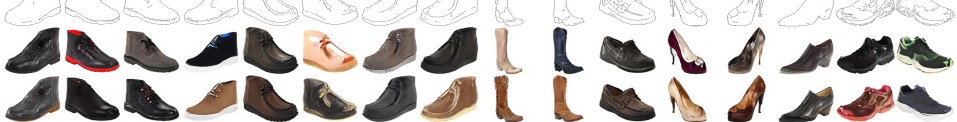

Figure 9: The generation example of MUNIT on the edge2shoes. Specifically, images at first two rows are source domain images and the others are translated images by MUNIT.

## A.3 EXPERIMENTS

### A.3.1 MAPS

The Maps dataset (Isola et al., 2017) contains 2194 aerial photo-map image pairs, with 1096 pairs for training and 1098 pairs for evaluation. For evaluation, we employ the metrics including RMSE and pixel accuracy with threshold $\delta$ ($\delta_1 = 5$ and $\delta_2 = 10$) suggested by GcGAN (Fu et al., 2019).

Table 6: Quantitative scores for Aerial photo→Map. ↓ indicates that the lower score is better and ↑ denotes that the higher score is better.

| Method | RMSE ↓ | acc%($\delta_1$) ↑ | acc%($\delta_2$) ↑ |
|---|---|---|---|
| GAN (baseline) | 33.2 | 19.3 | 42.0 |
| **GAN+MGC** | **28.9** | **38.6** | **61.8** |
| CycleGAN | 26.81 | 43.1 | 65.6 |
| **CycleGAN+MGC** | **26.61** | **44.7** | **66.2** |
| GcGAN-Mix | 27.98 | 42.8 | 64.6 |
| **GcGAN-Mix+MGC** | **26.55** | **44.7** | **66.5** |

The scores are reported in Table 6. Our MGC can significantly improve the accuracy to 38.6% and 61.8%, compared with the vanilla GAN, whose scores are 19.3% and 42.0% with the threshold of $\delta_1$ and $\delta_2$, respectively. Moreover, integrating our MGC constraint into CycleGAN and GcGAN can generate better translations than both individual ones. This further demonstrates the compatibility of the proposed mutual information method. Qualitative results are shown in Appendix A.7.1.

## A.4 EXPERIMENTAL DETAILS

We will public codes and experimental setting for the convenience of reproducing results in our paper.

### A.4.1 DIGITS

All digits images are resized to $32 \times 32$ resolution. Following Fu et al. (2019), the network details of this experiment are given in Table 7.

Table 7: The network details of digits translation tasks, where C = Feature channel, K = Kernel size, S = Stride size, Deconv/Conv = Deconvolutional/Convolutional layer and "channels" donotes the image channels of target domain, such as 1 for MNIST, 3 for MNIST-M.

| **Generator** | | | | |
|---|---|---|---|---|
| index | Layers | C | K | S |
| 1 | Conv + LeakyReLU | 64 | 4 | 2 |
| 2 | Conv + LeakyReLU | 128 | 4 | 2 |
| 3 | Conv + LeakyReLU | 128 | 3 | 1 |
| 4 | Conv + LeakyReLU | 128 | 3 | 1 |
| 5 | Deconv + LeakyReLU | 64 | 4 | 2 |
| 6 | Deconv + LeakyReLU | channels | 4 | 2 |
| 7 | Tanh | - | - | - |
| **Discriminator** | | | | |
| index | Layers | C | K | S |
| 1 | Conv + LeakyReLU | 64 | 4 | 2 |
| 2 | Conv + LeakyReLU | 128 | 4 | 2 |
| 3 | Conv + LeakyReLU | 256 | 4 | 2 |
| 4 | Conv + LeakyReLU | 512 | 4 | 2 |
| 5 | Conv | 512 | 4 | 2 |

Following all settings of the original models, the learning rate for generator and discriminator is 0.0002, the training epochs is 40000 and the batch size is 64.

### A.4.2 CITYSCAPES

All images are resized to $128 \times 128$ resolution. Following Zhu et al. (2017); Fu et al. (2019), the network details of this experiment are given in Table 8.

Table 8: The network details of digits translation tasks, where C = Feature channel, K = Kernel size, S = Stride size, Deconv/Conv = Deconvolutional/Convolutional layer and ResBlk = A residual block

| **Generator** | | | | |
|---|---|---|---|---|
| index | Layers | C | K | S |
| 1 | Conv + ReLU | 64 | 7 | 1 |
| 2 | Conv + ReLU | 128 | 3 | 2 |
| 3 | Conv + ReLU | 256 | 3 | 3 |
| 4-9 | ResBlk + ReLU | 256 | 3 | 1 |
| 10 | Deconv + ReLU | 128 | 3 | 2 |
| 11 | Deconv + ReLU | 64 | 3 | 2 |
| 12 | Conv | 3 | 7 | 1 |
| 13 | Tanh | - | - | - |
| **Discriminator** | | | | |
| index | Layers | C | K | S |
| 1 | Conv + LeakyReLU | 64 | 4 | 2 |
| 2 | Conv + LeakyReLU | 128 | 4 | 2 |
| 3 | Conv + LeakyReLU | 256 | 4 | 2 |
| 4 | Conv + LeakyReLU | 512 | 4 | 1 |
| 5 | Conv | 512 | 4 | 1 |

Following all settings of the original models, the learning rate for all generators and discriminators is 0.0002, the batch size is 1 and the training epochs for CUT is 400 and other models is 200.

### A.4.3 MAPS

All images are resized to $256 \times 256$ resolution. Following Zhu et al. (2017); Fu et al. (2019), the network details is similar to the details of Cityscape, but the generator contains 9 res-blocks for images with $256 \times 256$ resolution. Following all settings of the original models, the learning rate for all generators and discriminators is 0.0002, the batch size is 1 and the training epochs for CUT is 400 and other models is 200.

### A.4.4 STYLE TRANSFER

All settings are same with Maps A.4.3. The details of datasets as follows:
**selfie2anime**    This dataset is from U-GAT-IT Kim et al. (2019), which contains 3400 training images and 100 images for test.
**horse2zebra**    This dataset is from CycleGAN Zhu et al. (2017), whose training sets contains 1,067 horse images and 1,334 zebra images. The test set consists of 120 horse images and 140 zebra images.
**portrait2photo**    This dataset is from DRIT Lee et al. (2019), whose training sets contains 6,452 photo images and 1,811 portrait images. The test set consists of 751 photo images and 400 portrait images. Following all settings of the original models, the learning rate for all generators and discriminators is 0.0002 and the training epochs for CUT is 400 and other models is 200.

### A.5 ANALYSIS ON THE CAT2DOG DATASET

To analyze the performance of our MGC on geometry-variant datasets, we incorporate our MGC constraint into CycleGAN model and train it on the cat $\rightarrow$ dog dataset. The results are shown as Figure 10 , we can see that the trained translation model can successfully translate dog images at the top row to cat images and preserve the basic image content (*i.e.* locations of eyes, mouth, directions of faces), even if there are some changes of geometric structure. However, as images at the bottom row show, the translation model fails to translate the dog images to cat images in a meaningful way, as the mouth of dogs block the background but the mouth of cats do not, and so the translation model need to "imagine" some background area that be blocked, which needs us to propose more constraints.

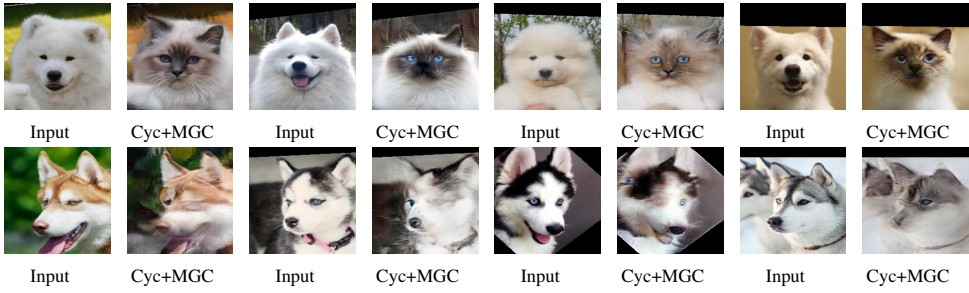

Figure 10: Qualitative results on a geometry-variant dataset, including Dog $\rightarrow$ Cat. Images at the top row are successful cases, while images at the bottom row are failure cases.

### A.6 FAILURE CASES ANALYSIS

Our MGC is a general geometry preservation constraint for I2I translation, aiming to reduce the randomness of color transformation in the translation. However, the style information can only be learned by GAN or other methods (e.g. U-GAT-IT). Limited by the absence of supervision signal, these methods sometimes recognize the semantic information wrongly, e.g. the background should be preserved in horse2zebra, but translation models sometimes mis-recognize some background to a horse in horse2zebra, and thus produce some zebra texture in the background in the translated image (best viewed at zoom level 200%). After coupling with our MGC, the background similar to the mis-recognized background is also likely be mistranslated (e.g. more background is mistranslated to texture of zebra). Although there are a small number of such cases occur in the translation process, the geometric details (e.g. sea or beach or sky) that are not mis-recognized can be better preserved with our MGC.

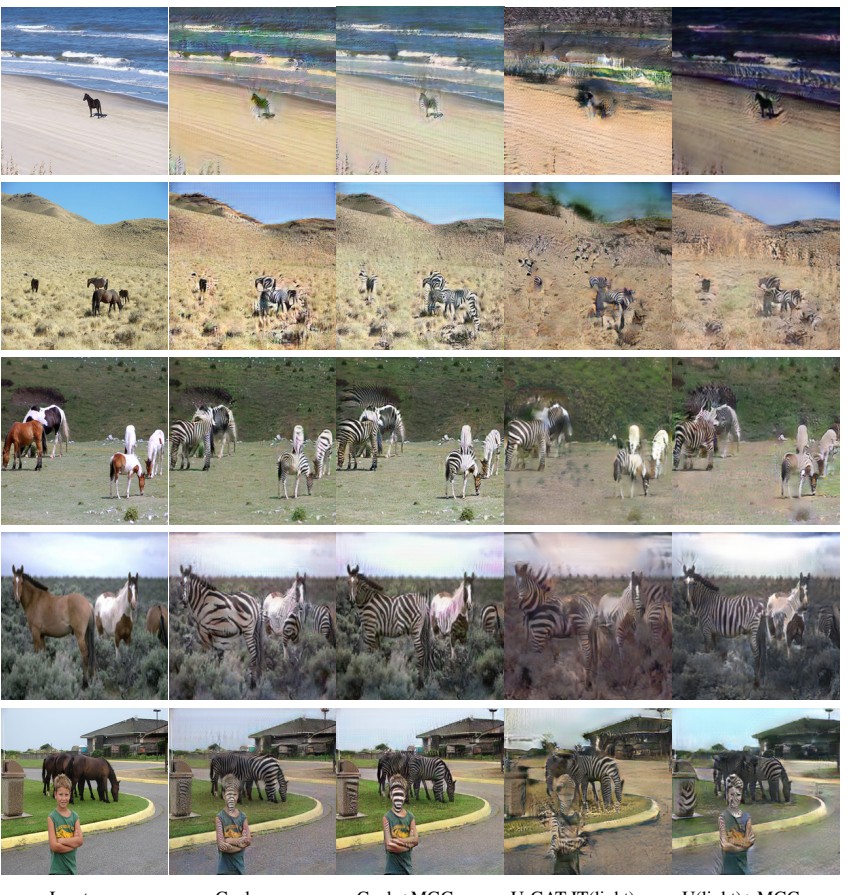

| Input | Cycle | Cycle+MGC | U-GAT-IT(light) | U(light)+ MGC |

Table 10: Falure cases on the horse2zebra dataset.

## A.7 GENERATED SAMPLES

### A.7.1 MAPS

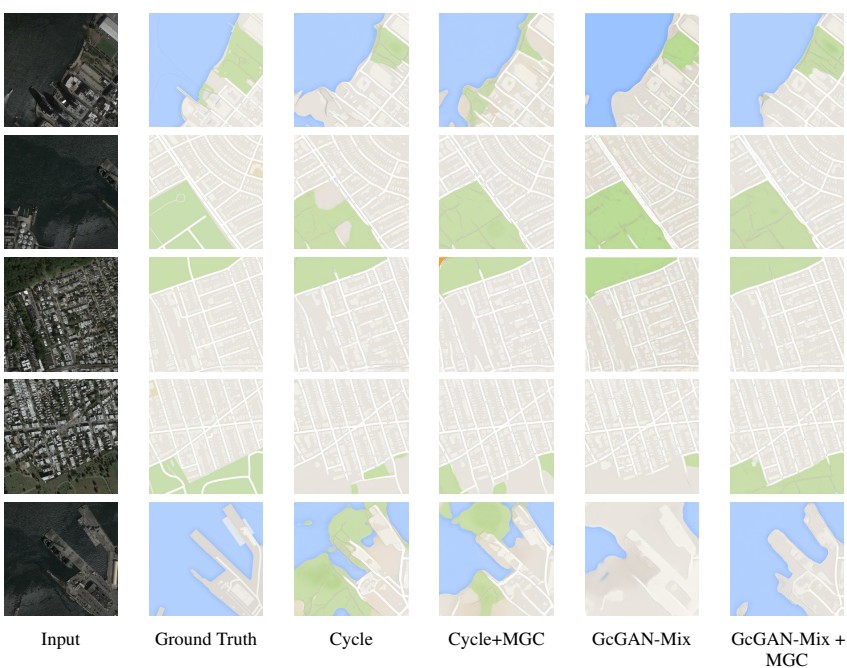

| Input | Ground Truth | Cycle | Cycle+MGC | GcGAN-Mix | GcGAN-Mix + MGC |

Table 11: Qualitative results on the Maps dataset.

### A.7.2 CITYSCAPES

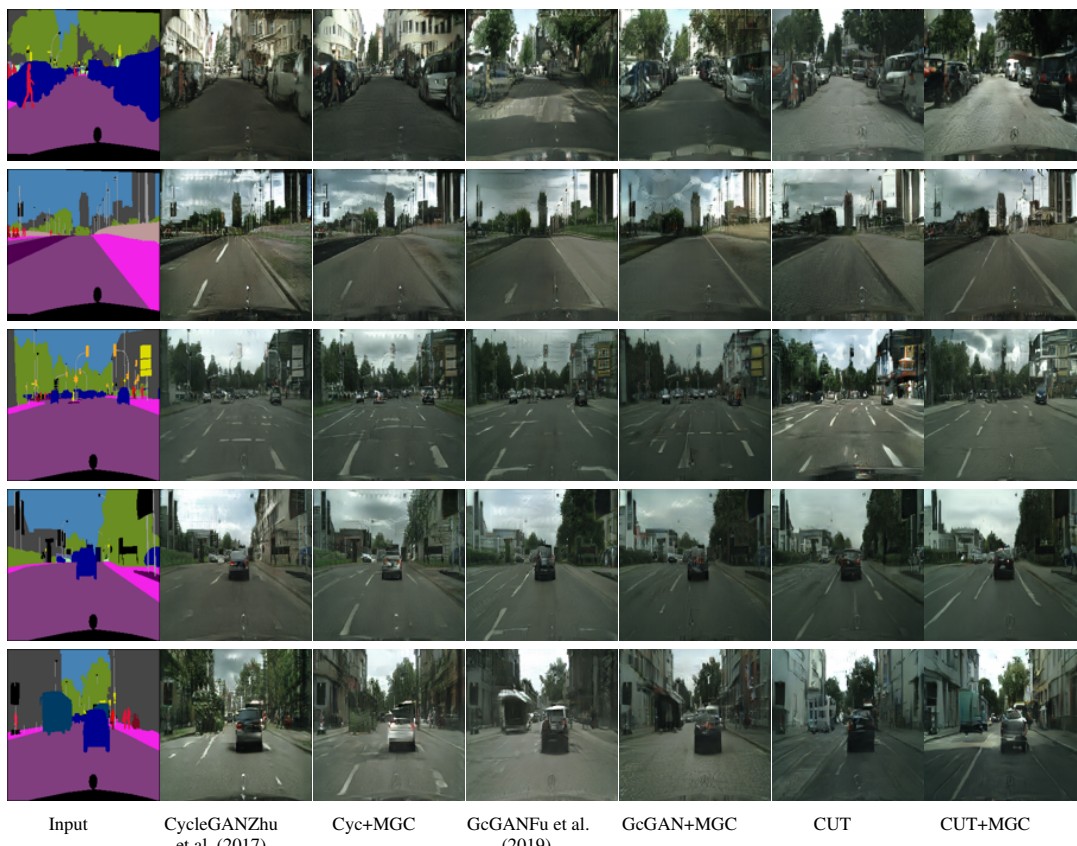

| Input | CycleGANZhu et al. (2017) | Cyc+MGC | GcGANFu et al. (2019) | GcGAN+MGC | CUT | CUT+MGC |

Table 12: Qualitative results on the Cityscape Dataset.

### A.7.3 STYLE TRANSFER

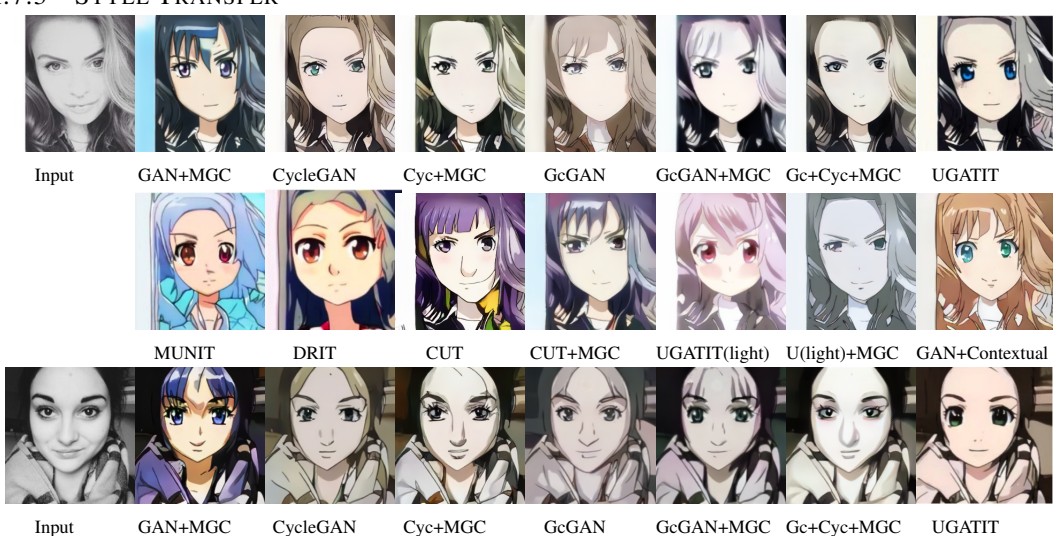

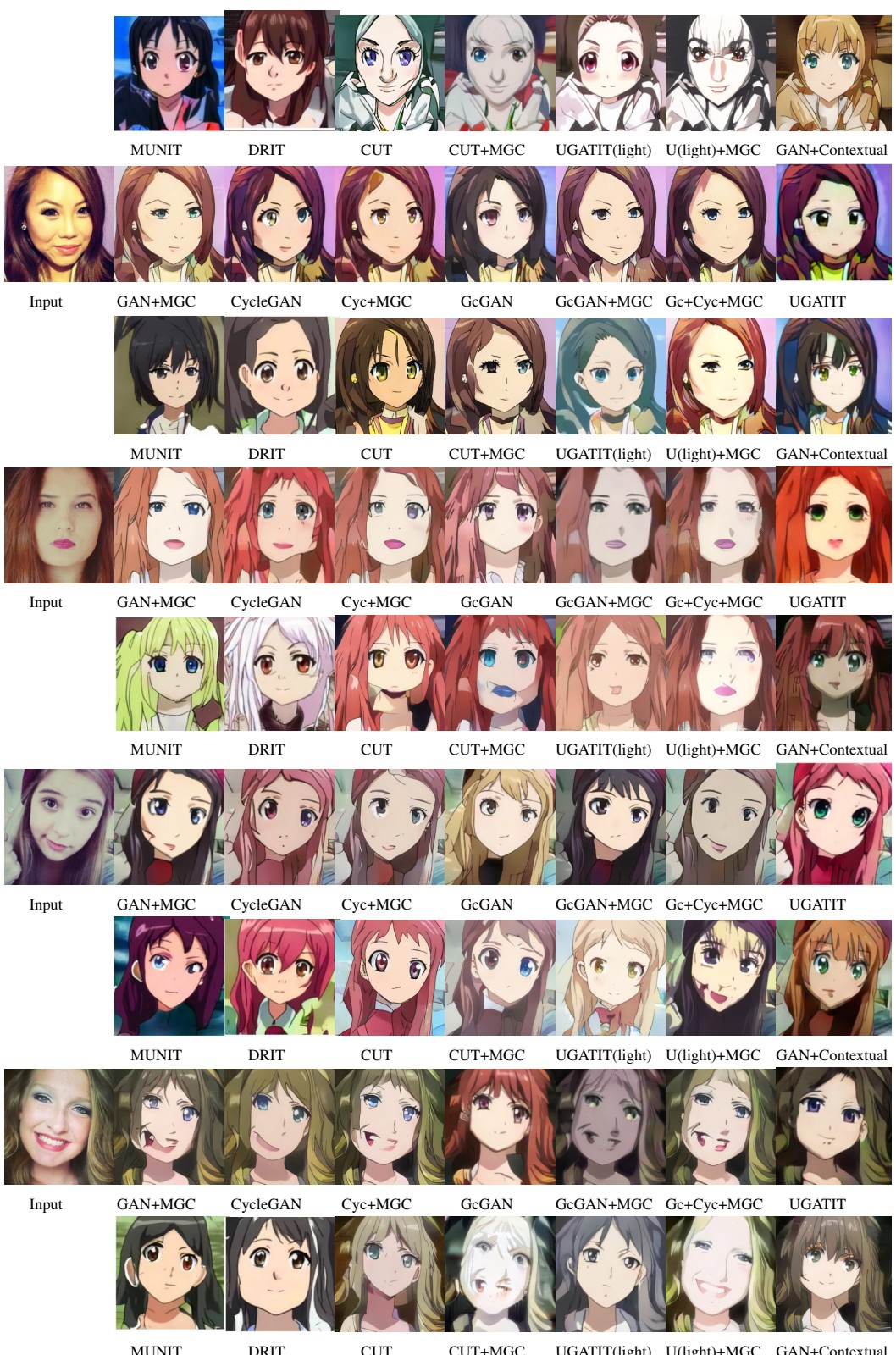

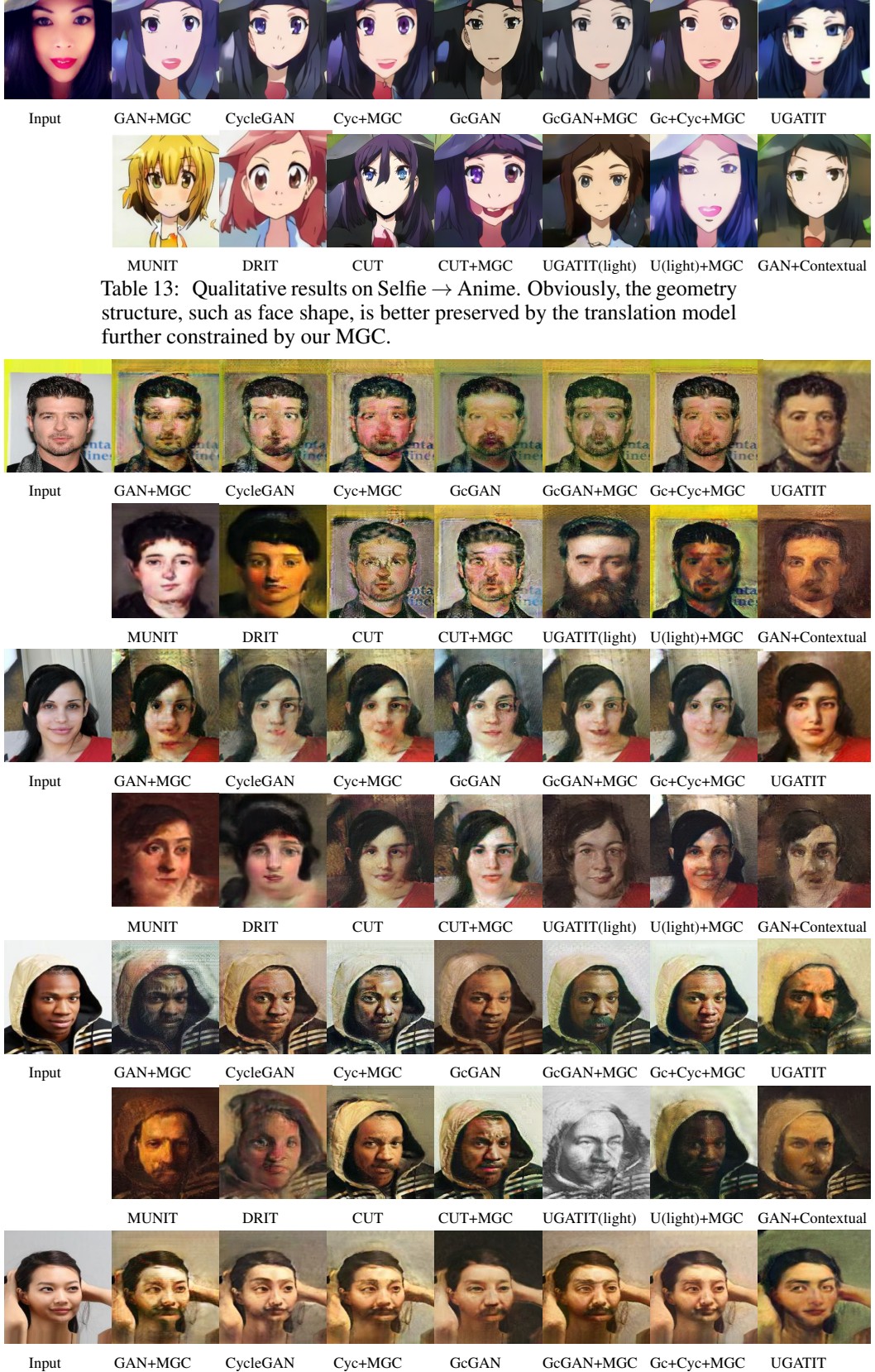

Table 13: Qualitative results on Selfie → Anime. Obviously, the geometry structure, such as face shape, is better preserved by the translation model further constrained by our MGC.

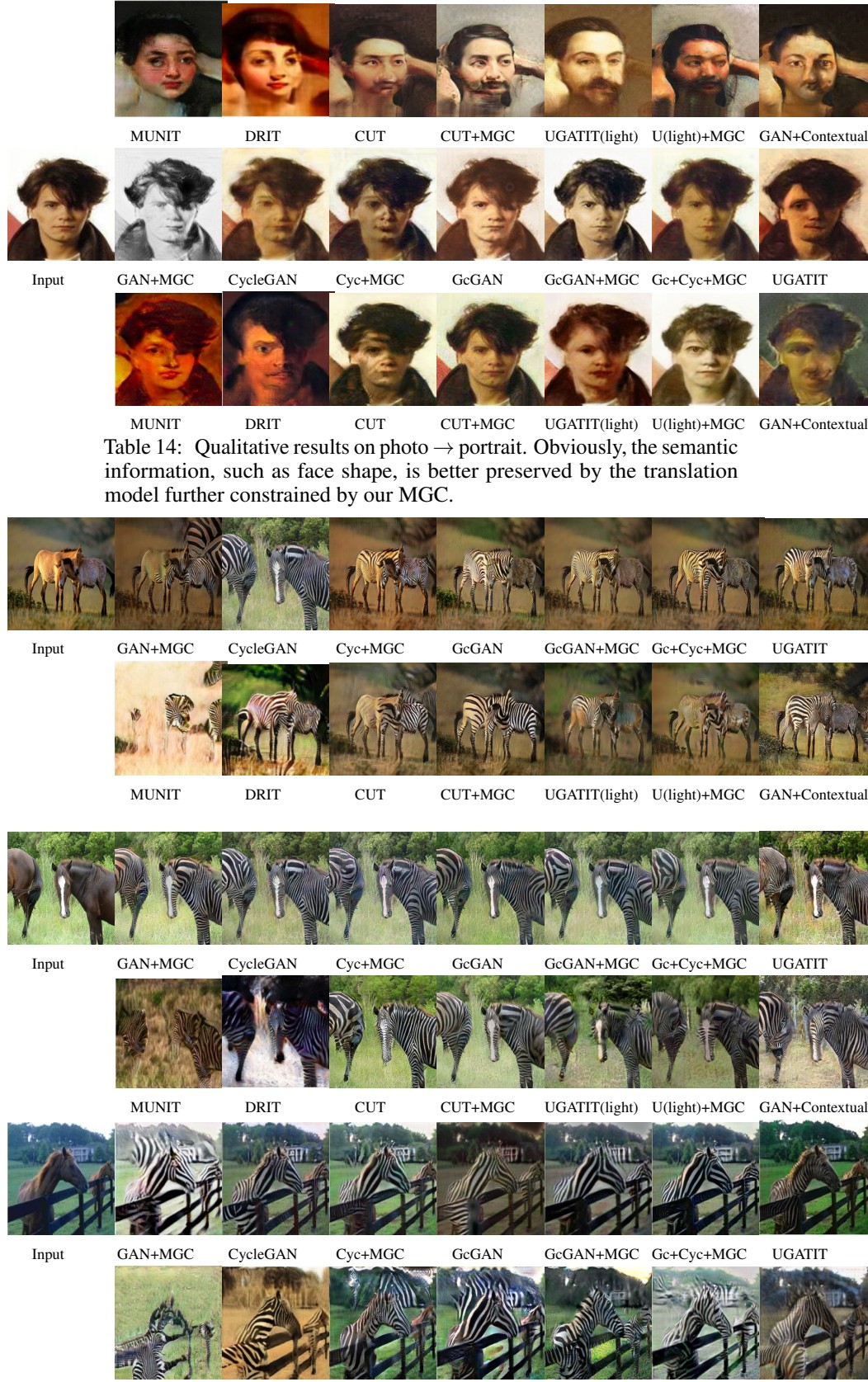

Table 14: Qualitative results on photo → portrait. Obviously, the semantic information, such as face shape, is better preserved by the translation model further constrained by our MGC.

Table 15: Qualitative results on Horse → Zebra. Obviously, the semantic information, such as background, is better preserved by the translation model further constrained by our MGC.

## A.7.4 DIGITS

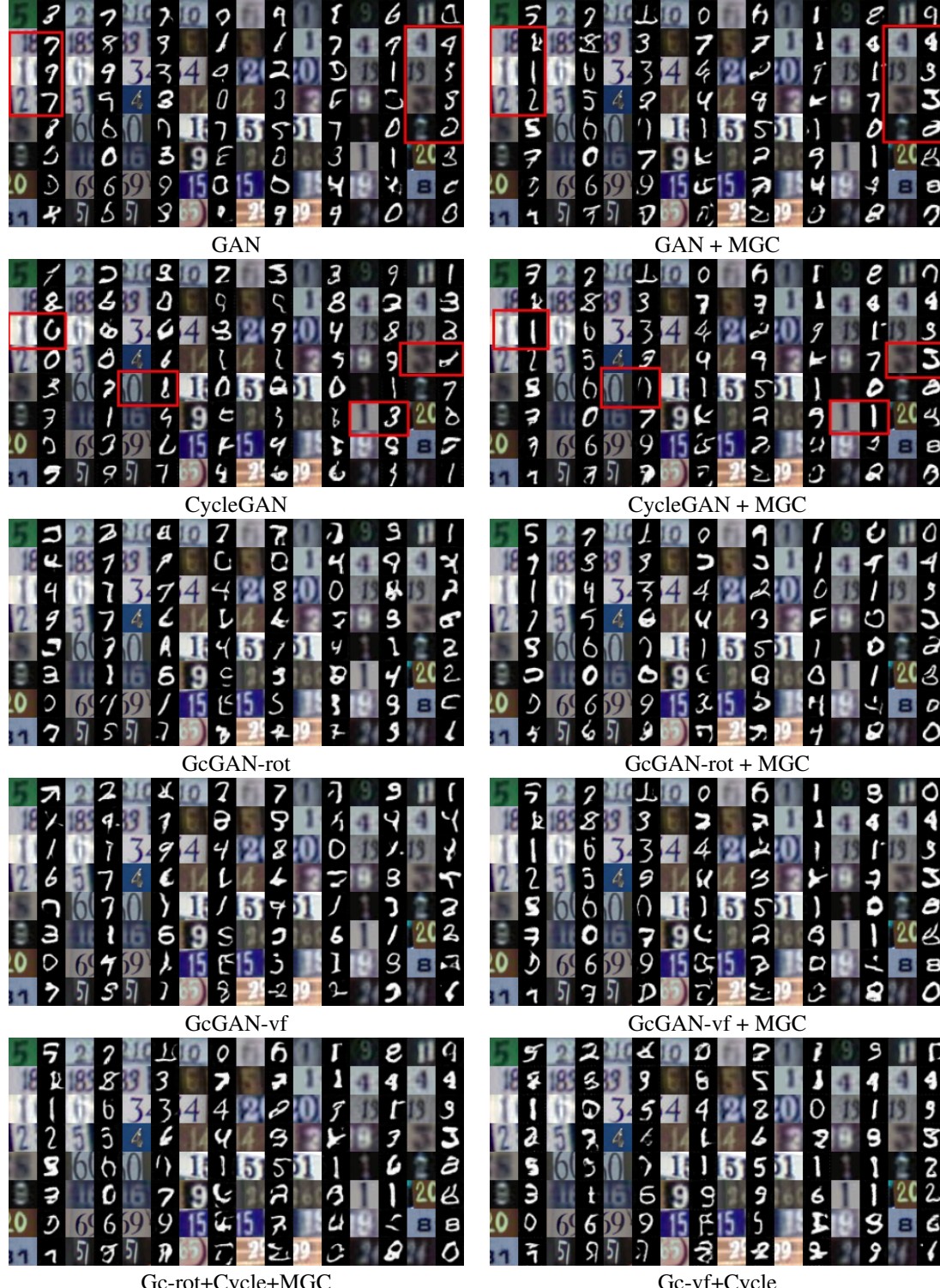

Table 16: Qualitative comparisons on SVHN→MNIST.

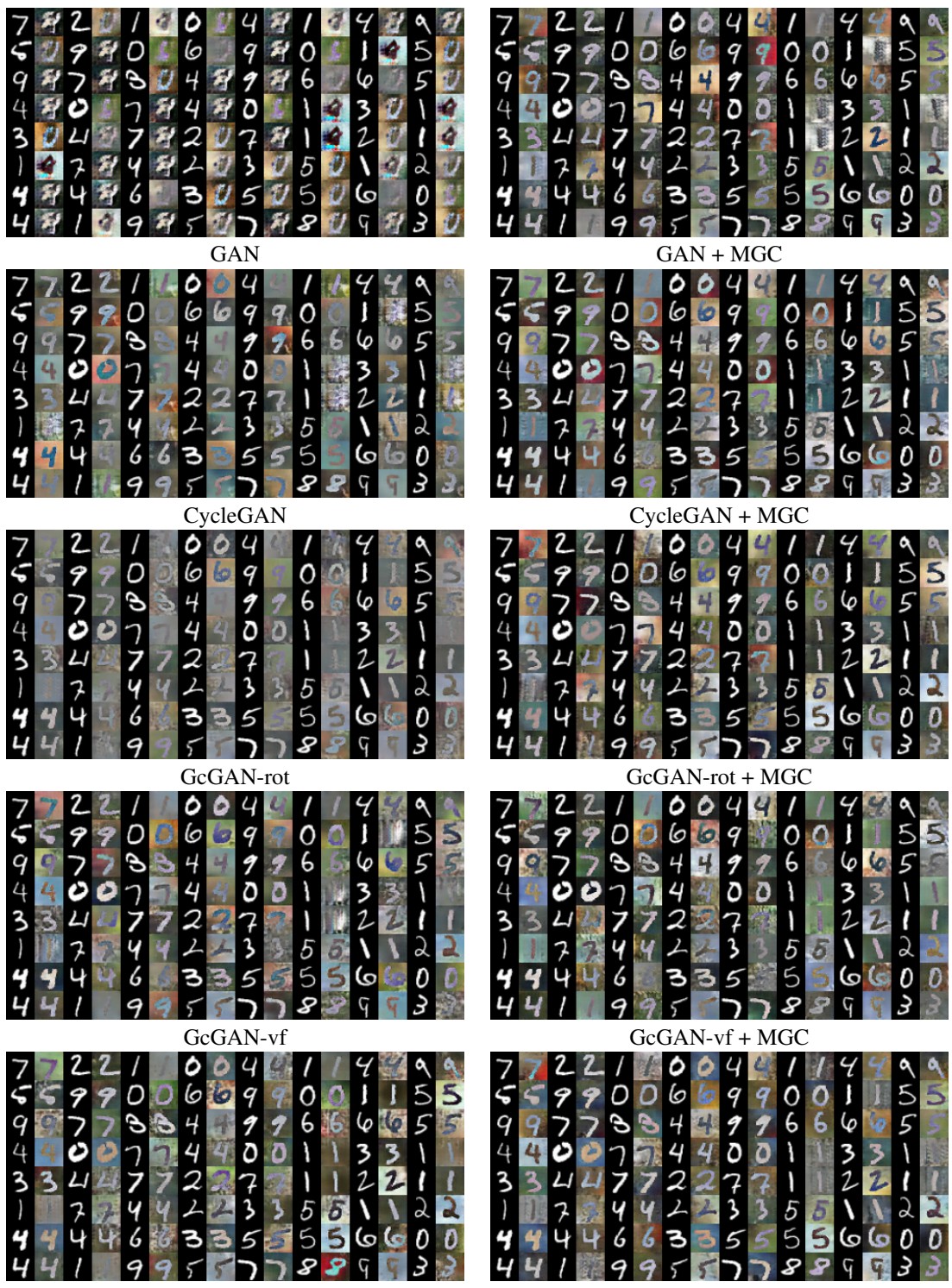

GAN

GAN + MGC

CycleGAN

CycleGAN + MGC

GcGAN-rot

GcGAN-rot + MGC

GcGAN-vf

GcGAN-vf + MGC

Gc-rot+Cycle+MGC

Gc-vf+Cycle

Table 17: Qualitative comparisons on MNIST→MNIST-M.

### A.7.5 ABLATION STUDY

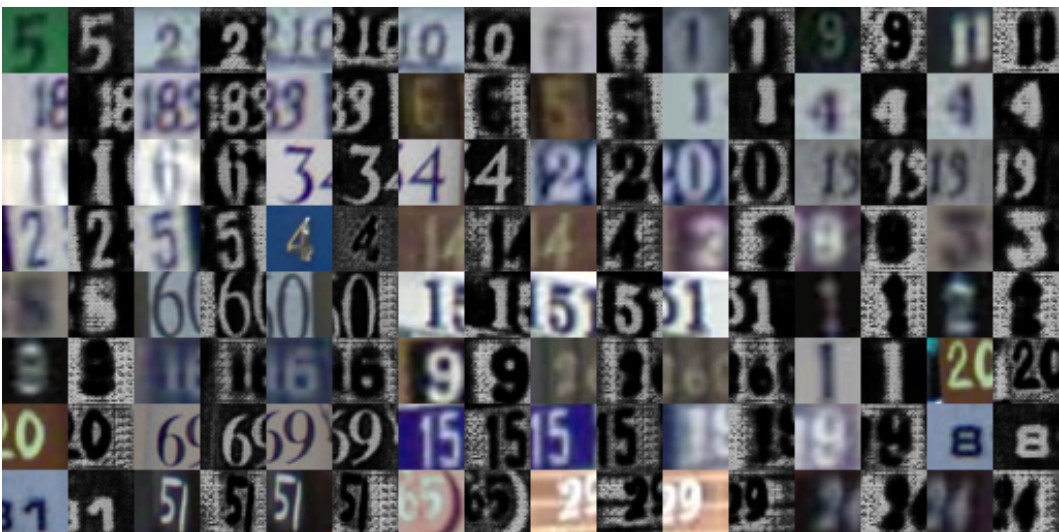

Figure 11: The overlarge $\lambda_{mgc}$ example on SVHN→MNIST.

An example of SVHN to MNIST translation when $\lambda_{mgc}$ is set to 25 is shown as Figure 11. The images are almost translated without any changes in geometry structures. However, the overlarge $\lambda_{mgc}$ causes the translation model neglect the style information from adversarial loss, resulting in some images with opposite color. This phenomenon indicates that our MGC has good performance on the preservation of geometry structure but should be appropriate with style information.

