# OpenReview forum: "Minimal Geometry-Distortion Constraint for Unsupervised Image-to-Image Translation"
_ICLR.cc/2021/Conference — Reject_

### Official Review · AnonReviewer2 · 2020-10-28
**Minimal Geometry-Distortion Constraint for Unsupervised Image-to-Image Translation**

**Rating:** 4
**Confidence:** 5

**Review:**

#####################################################

Summary:

The paper focuses on the geometry distortion problem of unsupervised image-to-image translation. To combat this issue, authors approach a new I2I constrain: Minimal Geometry-Distortion Constrain (MGC). In practice, performing estimation and maximization of MGC is challenge, then the paper provides an approximate representation of mutual information: relative Squared-loss Mutual Information (rSMI). To evaluate the proposed method, various kind of datasets are leveraged and compared to SOTA.

 #####################################################

Pros:

+ From both quantitative and qualitative results, the proposed obtains batter score compared to baselines.

+ A new mutual information is proposed and utilized for Image-to-image translation.

+ The paper conducts extensive experiment on the various kind of datasets.

+This paper is well-written and easy to understand.

  #####################################################

Cons:

-For me, the reason why the mutual information preserves the structural information  is unclear. I fail to understand why the proposed method  only focus on the geometry information and ignores the other (e.g. color, style and pattern).   If authors utilizes the pre-trained vgg (on ImageNet or celeba) to extract the feature of both the input and output image, and then use reconstruction loss to constrain the output images, which I think still be helpful to constrain the geometry and ignore the colour information. I would like the new mutual information objuective, but do not agree it is suitable for Image-to-image translation.

-The improved result (e.g. KID, Table 3) is weird. I think the proposed method is added directly to current frameworks. For example, using CycleGAN + MGC makes the model less freedom to generate the target domain, which probably results in less performance than CycleGAN.

-The most dataset maybe not suitable for the proposed method, since the  dataset except for the tiny experiment on digit are allowed to change the geometry and should be altered to adapt the target domain.  In fact more papers[1,2] give effort to change the geometry, since current methods is limited to change the geometry.

-I am not sure how authors select baselines. In the first paragraph of Experiment section, authors introduce the compared baseline in this paper. The following result (Table1, 2), however, present different ones (e.g. GcGAN, CoGAN, BiGAN/ALI, SimGAN, DistanceGAN), which makes me confused.   Besides, some datasets are performed on GcGAN-rot/vf (Table 1), but other one only contains GcGAN-rot (Table 2), and Table 3 show GcGAN (w/o rot/vf), which is weird for me.



[1] Cross-Domain Cascaded Deep Feature Translation, ECCV2020

[2] TransGaGa: Geometry-Aware Unsupervised Image-to-Image Translation, CVPR2019

---

> ### Author Response · Authors · 2020-11-17
> **Response to AnonReviewer2: Part1**
>
> Thanks for your constructive feedback. In the following, we would like to clarify our methods and the importance of unsupervised geometry-invariant I2I translation. After addressing these concerns, we think the contribution is clear, thus sincerely hope the reviewer to reconsider the decision.
> \
> $\textbf{The importance of geometry preservation in unsupervised I2I translation.}$\
> We understand the interest of the review on geometry variant I2I translation, e.g., transfer images between very different object categories, and we will add discussions about the recommended papers that give efforts to change the geometry. However, the geometry preservation method is also important in many unsupervised I2I translation tasks. Some unsupervised I2I translation applications that need geometry structures preservation are summarized below:
>
> 1. Domain adaptation: There are abundant literature on the usage of unsupervised I2I translation to improve the performance of Domain adaptation, such as CyCADA[1], I2I Adapt[2]. However, because of the absence of supervised signal in the target domain, translation models often change the geometry information of the source image, leading to changes of semantic information in the translated image (See examples at Figure 3 under digits experiments section in the main paper). The unsupervised geometry-invariant I2I translation can avoid this problem effectively, improving the performance of the adapted model.
>
> 2. Unsupervised image segmentation and generation based on segmentation: This is a popular and complex I2I translation task, but neglected by many I2I methods. By using our MGC, translation model can preserve the geometry structures better, and thus improve its performance on Cityscape segmentation dataset.
>
> 3. Style transfer applications: In many style transfer applications, the change of geometry structure may result in the change of semantic information, such as in the translation of satellite photos to Google maps in section A.4, portrait2photo, selfie2anime, simulation2real. In these tasks, previous methods failed to preserve the geometry structure of source images, leading to the mismatch between source and target domain images.
>
> Overall, there are many applications of unsupervised I2I translation require the geometry preservation between source and target images and our MGC is an effective constraint that can be used in these applications.
>
> $\textbf{The superiority of MGC over pre-trained VGG}$
> 1. Pre-trained VGG is hard to guarantee that the geometry structures are unchanged, while our MGC is a simple way to solve it.
>     - Essentially, the feature extracted from Pre-trained VGG is abundant, containing more than just the geometry information of image. However, because of the absence of supervised signal in the target domain and the weak interpretability of neural network, it is hard to justify which feature is relevant to geometry information. In some worse cases, the feature of geometry information is even entangled with style information which must change in the translation process. Although many methods are trying to disentangle features and interpret neurons, the additional methods for I2I tasks are still complex while our MGC is a simple way to preserve geometry information of images.
>     - Experimentally, Contextual Loss based [3] on Pre-trained VGG has been chosen as a baseline in our paper, which aims to reduce the distance of feature extracted from Pre-trained VGG between source and target images. In addition, Contextual Loss is a more recent and representative Pre-trained VGG based method, the performance of it is superior over the original popular content loss [8]. However, Figure 2 and qualitative results in Appendix A.7.3 show that the translation models with MGC can preserve geometric structures better than models with Contextual Loss.
> 2. Pre-trained VGG is computationally costly.
>     - The Pre-trained VGG-19 needs over 580MB additional memory, while our MGC only needs less than 1MB .
>     - The model with VGG-19 needs much longer training time than the model with MGC. Taking the horse2zebra as an example, the training time for the VGG-19 is 20 hours, while the training time for the model with MGC is only 10 hours, using a single 1080Ti.
> 3. Due to the distribution shift between ImageNet and the I2I dataset at hand [7], the pre-trained VGG model is not optimal for any given new I2I dataset.
>
> [to be continued]

---

> > ### Author Response · Authors · 2020-11-17
> > **Response to AnonReviewer2 : Part2**
> >
> > $\textbf{	Why can mutual information preserve the geometric structure of images?
> > }$\
> > As mentioned by AnonReviewer1, preserving the edges in images is a potential way to preserve geometric information of images, and we find that reducing the color transformation in the translation process can achieve it in a simple way.
> > Normally, the edges in an image can be regarded as the boundaries of different adjacent color regions. If we want the original edges to be preserved, and no new edges to be generated in the translation process, the adjacent color regions within boundaries should be transformed consistently. Taking a red square with black borders as an example, we can recognize the red color region as a square, because the edge with a square shape is generated by the red color region and the black color region as the borders. If the black color in the periphery is transformed into several different colors (e.g. blue and grey) randomly, the black region in the source image will be divided into several new regions (e.g. blue and grey) in the translated images. As a result, the edges generated by new adjacent color regions (red, blue and grey) are very likely to be irregular, distorting the square shape. Therefore, the randomness of color transformation in the translation process is harmful to preserving the edges and geometry structures in the images. In our paper, to reduce the randomness of color transformation in the translation process, we use mutual information to model the non-linear dependencies between colors in source and translated images. By maximizing the mutual information, the dependency between colors in source and translated images will become stronger, and the randomness of color transformation will be reduced. Therefore, the geometric structures can be better preserved. In the updated version of our paper, we add examples about how the randomness of color transformation existing in previous I2I method distorts the geometry structures and how our method reduces it to preserve geometry structure better in the translation process.
> > \
> > $\textbf{“Weird results because the model with MGC has less freedom”:
> > }$\
> > The evaluation metric KID is a widely used metric in I2I translation papers, such as U-GAT-IT [4], AG-GAN[5], FQ-GAN [6], and the comparison of different models is fair. In addition, the reviewer mentioned that CycleGAN with MGC has less freedom to generate target images, but in the same way, GAN with cycle constraint has less freedom to generate target images compared with GAN alone, the performance of CycleGAN is still higher than vanilla GAN. Therefore, the results in our paper are not weird and the performance of the model with a constraint can be improved as long as the constraint is reasonable. As qualitative results in A.7 and user study show, our MGC actually can preserve the geometry structures in source images, and thus the translated images are more visually appealing. To test how our constraint could possibly affect the diversity of generated images, we conducted an experiment on edge2shoes in Section A.2.2. The results show that our constraint does not hurt generation diversity.
> > \
> > $\textbf{	Baselines:
> > }$\
> > To evaluate the effectiveness and robustness of our MGC, we run experiments on three applications of I2I translation. The current baselines have their own advantages and disadvantages: some baselines perform well on one task but perform poorly on other tasks. For example, some style transfer methods do not perform well on unsupervised image segmentation. As such, following the current literature, we compare our methods with SOTA methods for each application, and the performance of all most all baselines are improved to different extents after incorporating with our MGC, which are sufficient to support our claim that MGC is a general constraint to preserve geometry structure at I2I translation.

---

> > > ### Author Response · Authors · 2020-11-17
> > > **Response to AnonReviewer2 : Part3**
> > >
> > > [1] Hoffman, Judy, et al. "Cycada: Cycle-consistent adversarial domain adaptation." International conference on machine learning. PMLR, 2018.\
> > > [2] Murez, Zak, et al. "Image to image translation for domain adaptation." Proceedings of the IEEE Conference on Computer Vision and Pattern Recognition. 2018.\
> > > [3] Mechrez, Roey, Itamar Talmi, and Lihi Zelnik-Manor. "The contextual loss for image transformation with non-aligned data." Proceedings of the European Conference on Computer Vision (ECCV). 2018.\
> > > [4] Kim, Junho, et al. "U-gat-it: unsupervised generative attentional networks with adaptive layer-instance normalization for image-to-image translation." ICLR 2020.\
> > > [5] Tang, Hao, et al. "Attention-guided generative adversarial networks for unsupervised image-to-image translation." 2019 International Joint Conference on Neural Networks (IJCNN). IEEE, 2019.\
> > > [6] Zhao, Yang, et al. "Feature Quantization Improves GAN Training." ICML 2020.\
> > > [7] Park, Taesung, et al. "Contrastive learning for unpaired image-to-image translation." ECCV 2020.\
> > > [8] Gatys, Leon A., Alexander S. Ecker, and Matthias Bethge. "Image style transfer using convolutional neural networks." CVPR 2016.
> > >
> > > $\textbf{Thanks for your attention and please let us know if we can address your concerns.}$

---

### Official Review · AnonReviewer3 · 2020-10-28
**The proposed constraint based on relative Squared Mutual Information (rSMI) would be useful in correcting geometric distortions in unsupervised image-to-image (I2I) translation tasks.**

**Rating:** 7
**Confidence:** 4

**Review:**

This work introduces the geometry-distortion constraint (MGC) for mitigating undesired geometric distortions that may often occur in the current unsupervised image-to-image (I2I) translation approaches. The MGC is formulated using the Mutual Information (MI) between intensity values from original image and translated image. Authors claimed that the maximization of MI enables for suppressing the randomness of color transformation, leading to geometric-distortion free results. To integrate the MI into GAN based deep networks, authors leveraged the relative Squared Mutual Information (rSMI) that is differential, proposed in (Sugiyama et al., 2013). Experiments support the effectiveness of the proposed MGC in various image translation tasks, including digit translation, image-to-parsing, and style transfer.

* Pros
1) The MGC based on rSMI seems to be beneficial to several I2I translation tasks.
2) Detailed ablation studies on various tasks are provided.

* Cons
1) The proposed MGC is applicable only for some limited tasks, where geometric layouts are strongly preserved. Namely, deforming objects in a geometric fashion does not belong to such cases.
2) For general scenes with multiple similar objects, this constraint may lead to severe distortions. For instance, the results in the last row of Figure 4 demonstrate that the proposed MGC yields even worse results than original methods. Failure cases and analysis on the limitation of the proposed constraint should be provided carefully.

Minor comments
1) Appendix A.1 seems to be from existing works, and thus references would be needed.
2) A.3 has no contents.
3) What is Gc+Cycle+MGC in Table 3?
4) What is GAN+Contextual in Table 3?
5) Setting lambda_mgc is described below. Please specify how to choose lambda_mgc in more details.
'A practical strategy of choosing lambda_mgc is to find the largest lambda_mgc with normal style information using binary search.'

---

> ### Author Response · Authors · 2020-11-17
> **Response to AnonReviewer3**
>
> We appreciate the encouraging feedback and constructive feedback from the reviewer. In the following, we will address your concerns.
> \
> $\textbf{Cons 1 “Limited applications”:
> }$\
> Our MGC aims to preserve the geometry structure in the source images in I2I tasks. Such preservation of geometry structure is often demanded in various applications, and we have summarized several examples below:
> 1. Domain adaptation: There are abundant literature on the usage of unsupervised I2I translation to improve the performance of Domain adaptation, such as CyCADA[1], I2I Adapt[2]. However, because of the absence of supervised signal in the target domain, translation models often change the geometry information of the source image, leading to changes of semantic information in the translated image (See examples at Figure 3 under digits experiments section in the main paper). The unsupervised geometry-invariant I2I translation can avoid this problem effectively, improving the performance of the adapted model.
> 2.Unsupervised image segmentation and generation based on segmentation: This is a popular and complex I2I translation task, but neglected by many I2I methods. By using our MGC, translation model can preserve the geometry structures better, and thus improve its performance on Cityscape segmentation dataset.
> 3.Style transfer applications: In many style transfer applications, the change of geometry structure may result in the change of semantic information, such as in the translation of satellite photos to Google maps in section A.4, portrait2photo, selfie2anime, simulation2real. In these tasks, previous methods fail to preserve the geometry structure of source images, leading to the mismatch between source and target domain images.
> Overall, there are many applications of unsupervised I2I translation require the geometry preservation between source and target images, and our MGC is an effective constraint that can be used in these applications.
>
> $\textbf{Cons 2:
> }$\
> Thanks for your suggestion. Our MGC aims to reduce the randomness of color transformation in the translation, preserving the geometry structures of images, but what part of image (e.g. horse) should be translated is learnt by GAN or disentangle methods (e.g. MUNIT, DRIT). Limited by the absence of supervision signal, these methods sometimes recognize the semantic information wrongly, e.g. misrecognize the background to a horse in horse2zebra, and thus has some zebra texture in the background in the translated image (zoom in can see it). After coupling with our MGC, the background similar to the misrecognized background will also be mistranslated (e.g. more background is mistranslated to the texture of zebra). Although there are a small number of cases occur in the translation process, but the geometry details that is not misrecognized can be still preserved (e.g. the fence of the first horse image, and the white patch in the horse head in the second image). In general, the methods with simple constraints (e.g. Gc and Cycle) are less likely to misrecognize information of images than those larger and more complex models. As such, a small 9-layer res-net model (45M) with Gc+Cycle constraints and our MGC perform SOTA performance over almost all datasets and obtain the highest score in the user study compared again with other large models (U-GAT-IT:134M; DRIT 65M; MUNIT 46M). In the updated version, we will add the failure case analysis in Appendix A.6.
>
> [to be continued]

---

> > ### Author Response · Authors · 2020-11-17
> > **Response to AnonReviewer3 : Part2**
> >
> >
> > $\textbf{Minor Comments 1:}$\
> > Thanks for your suggestion, we will add the references for it in the updated version of the paper.
> > \
> > $\textbf{Minor Comments 2:}$\
> > Thanks for your suggestion, we will correct it in the updated version of the paper.
> > \
> > $\textbf{Minor Comments 3:}$\
> > Gc+Cycle+MGC is the translation model trained with Gc(rot), cycle and MGC constraints, this model with only 45 trainable parameters can achieve higher performance than the models with larger size (e.g. U-GAT-IT:134M; DRIT 65M; MUNIT 46M) at User Study.
> > \
> > $\textbf{Minor Comments 4:
> > }$\
> > GAN+ Contextual is the model trained with GAN objective function and the Contextual loss [3] which aims to reduce the distance between the source and translated images on the feature space of a pre-trained VGG network. The Contextual loss has a better performance the popular content loss at I2I translation but still perform worse than our MGC, showing that our MGC has the superiority over the loss based on pre-trained VGG.
> > \
> > $\textbf{Minor Comments 5:
> > }$\
> > Normally, we set the $\lambda_{mgc}$ as 5 (effective for most style transfer dataset). If the style of translated images is consistent with that of real target domain images, we can try to increase $\lambda_{mgc}$ as 10. If the style of translated images is significantly inconsistent with target domain style, the $ \lambda_{mgc}$ can be set to smaller values such as 2 or 3.
> >
> > [1] Hoffman, Judy, et al. "Cycada: Cycle-consistent adversarial domain adaptation." International conference on machine learning. PMLR, 2018.
> > [2] Murez, Zak, et al. "Image to image translation for domain adaptation." Proceedings of the IEEE Conference on Computer Vision and Pattern Recognition. 2018.
> > [3] Mechrez, Roey, Itamar Talmi, and Lihi Zelnik-Manor. "The contextual loss for image transformation with non-aligned data." Proceedings of the European Conference on Computer Vision (ECCV). 2018.
> >
> > $\textbf{Please feel free to let us know if you still have some concerns or questions.}$

---

### Official Review · AnonReviewer1 · 2020-10-29
**Needs better exposition**

**Rating:** 4
**Confidence:** 4

**Review:**

This paper presents an image to image translation framework that addresses the problem of preserving geometric details. Specifically, the goal is for the framework to translate the color details, however preserve the structural cues. To this end, the main idea is to derive a mutual-information constraint between the pixel colors of the input and translations, which is added as a regularization in the standard adversarial GAN loss. Experiments are provided on several examples and show some promise.

Pros:
1. The idea of preserving structural information in I2I translations is interesting.
2. Some of the translated images look appealing.

Cons:
1. It is unclear to me how precisely is the paper tying the geometry-preservation with mutual information? Aren't we supposed to capture the higher-order gradients to preserve the structure (for example, the edges in the images, and such)? However, what the paper is suggesting is to compute the relative squared-loss mutual information over all pixels. How will this preserve the geometry?

2. The quantitative results show only very marginal improvements over other methods. The qualitative results are also not significantly different from say cycle-GAN (in quality). e.g. Figure 4 U(light) + MGC on the zebra image.

3. The proposed approach appears to be very similar to methods that attempt to minimize the perceptual loss; such as for example "Generating Images with Perceptual Similarity Metrics based on Deep Networks, Dosovitskiy and Brox, NIPS 2016". It would be good to contrast the approach to such methods. There are several recent works in this area that the paper could compare to as well.

Minor comments:
a. The paper could benefit from better organization, and thorough polishing.
b. After (1), it is said that "it is unclear how we can backpropagate through the histogram". There are standard methods to do that like computing a soft-histogram.
c. What is P_{V^Y'} just before (2)? I think it must be P_{V^\hat{Y}}.
d. How precisely is the E_{\beta S_i + (1-\beta)Q_i} characterized in (4)? How do you sample from this distribution?
e. It is unclear why it is called minimal geometry distortion constraint. It is more of a regularization, and technically there is no specific geometric information in the constraint. It is more of enforcing perceptuality.

Overall, the contribution of the paper is unclear. The technical presentation does not seem to match what the contribution is claimed to be. Experiments show very marginal benefits and needs comparison with more recent and relevant methods.

---

> ### Author Response · Authors · 2020-11-17
> **Response to AnonReviewer1: Part1**
>
>
> Thanks for your valuable feedback, which could help us explain our contributions more clearly. In the following, we will clarify our contribution and Cons you mentioned. After addressing these concerns, we think the motivation and presentation is clear, thus sincerely hope the reviewer to reconsider the decision.
> \
> $\textbf{Contributions and “Cons 1”：}: $ \
> As you mentioned, a good way to preserve the geometry structure of an image is to capture and preserve the edges in this image. Our paper proposes a simple way, i.e. reducing the randomness of color transformation. Normally, the edges in an image can be regarded as the boundaries of different adjacent color regions. If we want the edges to be preserved, and no new edges to be generated in the translation process, the adjacent color regions within boundaries should be transformed consistently. Taking a red square with black borders as an example, we can recognize the red color region as a square, because the edge with a square shape is generated by the red color region and the black color region as the borders. If the black color in the periphery is transformed into several different colors (e.g. blue and grey) randomly, the black region in the source image will be divided into several new regions (e.g. blue and grey) in the translated images. As a result, the edges generated by new adjacent color regions (red, blue and grey) are very likely to be irregular, distorting the square shape. Therefore, the randomness of color transformation in the translation process is harmful for preserving the edges and geometry structures in the images. In our paper, to reduce the randomness of color transformation in the translation process, we use mutual information to model the non-linear dependencies between colors in source and translated images. By maximizing the mutual information, the dependency between colors in source and translated images will become stronger, and the randomness of color transformation will be reduced. Therefore, the geometric structures can be better preserved. In the updated version of our paper, we add examples (in Introduction) about how the randomness of color transformation existing in previous I2I method distorts the geometry structures and how our method reduces it to preserve geometry structure better in the translation process.
> \
> $\textbf{Cons2 “Marginal improvement and more comparison”:}$\
> First, we have compared with many recently published methods, such as CUT [3], U-GAT-IT [4] and CycleGAN [5]. Secondly, we apply our MGC into almost all popular I2I frameworks, and after incorporating with our MGC, the performances of these models are consistently improved in a wide range of unsupervised geometry-invariant I2I translation applications. Specifically,
> 1.	For the difficult I2I translation dataset SVHN$\rightarrow$MNIST in digits experiments, the average accuracy of models with our MGC is even improved 36\% compared to the original models.
> 2.	In style transfer tasks, a small 9-layer res-net model (45M) with Gc+Cycle constraints and our MGC achieves SOTA performance over almost all datasets, and obtain the highest score in the user study compared again with other large models (U-GAT-IT:134M; DRIT: 65M; MUNIT: 46M). To avoid the concern of cherry-picking, the images in the user study are from qualitative results of U-GAT-IT. Thus, we believe these evidences are sufficient to verify the effectiveness and robustness of our MGC.
>
>
> $\textbf{Cons3 Concern about perceptual loss:
> }$\
> We will cite the recommended paper and add discussions about the difference between our method and perceptual loss. As the clarification of our motivation and method in the response for “Cons1”, our method aims to increase the non-linear dependencies of color in the source image and the translated images using mutual information, while perceptual loss reduces the distance in a feature space obtained by pre-trained networks.
> In our paper, we have compared the performance again with the Contextual loss [1] which is similar and superior to perceptual loss or the original content loss [2] in I2I task. The Contextual loss is a more recent method based on perceptual loss, and it reduces the cosine distance on the feature extracted from a pre-trained VGG network. The results in Table 3 of our paper shows that the performance of GAN+Contextual is significantly worse than GAN+MGC. In addition, as Figure2 and qualitative results in Appendix show, the translated images from GAN+Contextual model have significant geometry distortion, which means that perceptual loss alone is not able to guarantee the consistency of geometry structure. \
> \
> [to be continued]

---

> > ### Author Response · Authors · 2020-11-17
> > **Response to AnonReviewer1: Part2**
> >
> > $\textbf{Minor Comments a: }$\
> > Thanks for your suggestion, we add more examples and clarification in the rebuttal revision.
> > \
> > $\textbf{Minor Comments b:}$\
> > Sorry for raising your misunderstanding, we will replace it with “Next, we will introduce how we estimate the mutual information between pixels from two domain images and backpropagate it to optimize parameters in the translation network.”We will add the discussion about soft-hist methods such as kernel density estimation in Related Work. Specifically, we base our mutual information estimation method on SMI, as it does not require density estimation, but directly estimate the density ratio instead, and the proposed rSMI resolves the numeric instability problem of SMI.
> > \
> > $\textbf{Minor Comments c:
> > }$\
> > Thanks, we will correct it in the rebuttal revision.
> > \
> > $\textbf{Minor Comments d:
> > }$\
> > $S_i$ and $Q_i$ are the Kronecker product of marginal distributions and the joint distribution respectively,
> > i.e. S_i = P_{V^{x_i}} \otimes P_{V^{\hat{y}_i}} and Q_i=P_{(V^{x_i},V^{\hat{y}_i})}.  E_{\beta S_i + (1-\beta)Q_i} is characterized as E_{\beta S_i + (1-\beta)Q_i} = \beta E_{S_i} + (1-\beta)E_{Q_i}, where $S_i$ is sampled from marginal distributions and $Q_i$ is sampled from the joint distribution. After derivation, $E_{S_i}$ can be estimated by kernel methods and becomes $(KK^T)\circ(LL^T)$ in $\hat H$, while $E_{Q_i}$ becomes $(K\circ L)(K\circ L)^T$ in $\hat H$.
> > \
> > $\textbf{Minor Comments e:
> > }$\
> > Same with the response for Cons 1.
> >
> >
> > $\textbf{Thanks for your attention and please let us know if we can address your concerns.}$
> >
> > [1] Roey Mechrez eg. The contextual loss for image transformation with non-aligned data. In ECCV 2018.
> > \
> > [2] Gatys, Leon A., Alexander S. Ecker, and Matthias Bethge. "Image style transfer using convolutional neural networks." CVPR 2016.
> > \
> > [3] Park, Taesung, et al. "Contrastive learning for unpaired image-to-image translation." ECCV 2020
> > \
> > [4] Kim, Junho, et al. "U-gat-it: unsupervised generative attentional networks with adaptive layer-instance normalization for image-to-image translation." ICLR 2020.
> > \
> > [5] Zhu, Jun-Yan, et al. "Unpaired image-to-image translation using cycle-consistent adversarial networks." CVPR 2017.

---

### Official Review · AnonReviewer5 · 2020-11-04
**Novel idea about geometry structure constraint and thorough experiments**

**Rating:** 7
**Confidence:** 3

**Review:**

This paper presents a geometry-distortion constraint for the unsupervised image-to-image translation for a better structural similarity between the source and the target, which is deducted from the pixel correlation. The experiments on multiple GAN frameworks and datasets show its effectiveness in reducing the shape distortions in generated images.

## Pros

- This paper is well-written and easy to follow. The main paper is well-organized in describing the problem and proposed methods. And the appendix provides more interesting details.
- The experiments are quite thorough. This paper includes almost all well-known unsupervised image-to-image translation works as baselines and show significant improvements on them. The sensitivity study clearly shows the influence of mutual information by changing its weights.
- The idea of using pixel-wise geometric in-variance to constrain the shape distortion is quite novel.

## Cons
- In some datasets, simply maintaining the shape similarity doesn't always mean visual-appealing results. Sometimes it leads to a weird mixture of styles&structures of both domains for some datasets, e.g., selfie2ainme. I would be appreciated if the author can discuss more the limitations of this method and extends the analysis in A.6.
- A minor problem: it might not be a good choice to run experiments on edge2shoes as in Figure 7, since there is actually no shape distortion between the source and target (ditto for Cityscape, Maps). Besides, the figure annotation is not clear: which row is MUNIT? Which row is MUNIT+MGC?
- More implementation details are needed: epochs, learning rate, weight $\lambda_{mgc}$, whether trained from scratch, etc. for these datasets.

---

> ### Author Response · Authors · 2020-11-17
> **Response to AnonReviewer5**
>
> We appreciate the encouraging feedback from the reviewer. In the following, we would like to clarify the three Cons:
> \
> $\textbf{Cons1 "Failure Cases"}: $\
> Thanks for your suggestion. our MGC aims to reduce the randomness of color transformation in the translation, preserving the geometry structures of images, but the style information is learned from GAN or disentangle methods. Sometimes, the $\lambda_{mgc}$ is too small to balance the style and geometry information, leading to a small number of weird mixture of styles\&structures. In the updated version, we will add the failure case analysis in Appendix A.6.
> \
> $\textbf{Cons2 “experiments on edge2shoes”}:$ \
> Firstly, it is necessary to perform experiments on edge2shoes/cityscape/maps datasets as both the source and target domains contain more than one geometry structure. For example, edge2shoes contains different type of shoes with various shapes. As such, GAN alone methods still suffer from the geometry distortion problem, for example, translating one type of edge shoes to another type of real shoes. Secondly, aiming to show the influence of our MGC on the generation diversity, i.e. the ability of generating diverse target domain images given one source domain image and different latent variables z, we run the experiments on the edge2shoes dataset, where one edge can be corresponding to several shoes with different colors. Following the experimental settings of MUNIT, we calculate the LPIPS score (higher LPIPS score denote the better generation diversity). The results in A.2.2 show that our MGC does not have negative influence on the generation diversity. Finally, we add the qualitative results of MUNIT on the edge2shoes dataset in the updated version of our paper and add more detailed descriptions of the figure. Specifically, images in the first two rows are from source domain, others are translated image.\
> $\textbf{Cons3 “details of experiments”}:$
> \
> Thanks for your suggestions. In our experiments, we add our MGC into several popular I2I methods to evaluate the effectiveness and robustness of MGC, and thus follow their original experimental settings (epochs, learning rate). For a fair comparison, we also train all models from scratch. Besides,$ \lambda_{mgc} $is given at experimental descriptions, e.g. $\lambda_{mgc}$ is 5 in Style Transfer tasks. In the updated version of paper, we will add more details about experiments at Experiment section and the Appendix and release codes and experimental settings for the convenience of reproducing results in our paper.
>
> $\textbf{Please feel free to let us know if you still have some concerns or questions.}$

---

### Author Response · Authors · 2020-11-21
**Revision Uploaded**

We have uploaded a revision of our paper, and the modified part is typed in blue.

$\textbf{We summarize the modifications here}$:
1.	We add an illustration in $\textbf{Figure 1}$ to explain how the randomness of color transformation in the translation process harms to the geometry preservation.
2.	We add the discussion about unsupervised geometry-variant translation in the Related Work, following R#2's suggestions.
3.	We add the discussion about the perceptual loss in the Related Work, following R#1's suggestions.
4.	We add the reference about the process of solving our proposed MGC in the A.1, following R#3's suggestions.
5.	We add the Failure Cases Analysis in Appendix A.6, following R#5 and R3's suggestions.
6.	We add more experimental details in Appendix A.4, following R#5’s suggestions.
7.	We add the generation examples about MUNIT on edge2shoes dataset in Figure 9, following R#5’s suggestions.
8.	We refine the description of experimental settings and baselines, following R#2’s suggestions.
9.	We correct typos mentioned by R#1 and R#3.

$\textbf{We thank all the anonymous reviewers for their valuable suggestions and look forward to their further comments!}$

---

### Decision · Program_Chairs · 2021-01-07
**Final Decision**

**Decision:**

Reject

**Comment:**

This paper deals with unsupervised image-to-image translation and proposed a geometric constrains for better structural similarity between the source and the target. Experiments are done using multiple GAN frameworks and demonstrate reduction in distortions in the generated images.

The reviewers appreciated the contributions, but were overall not very enthusiastic about the paper, with two rejection recommendations. In particular, the criticism regarded
- limited applicability; shape similarity does not always translate into a good visual result; scenes with multiple similar objects might be severely distorted
- some results show a strange mixture of styles
- missing implementation details
- only small quantitative improvement
- similarity to prior works on perceptual loss
- lack of clarity about the use of mutual information for geometry preservation, and implementation details
- unconvincing baselines

The authors provided an extensive rebuttal addressing some of the above comments. However, many of the doubts remained because of which we believe the paper cannot be accepted.